# Instruct and Extract: Instruction Tuning for On-Demand Information Extraction

**Yizhu Jiao, Ming Zhong, Sha Li, Ruining Zhao, Siru Ouyang, Heng Ji, Jiawei Han**
University of Illinois Urbana-Champaign
`yizhuj2@illinois.edu`

## Abstract

Large language models with instruction-following capabilities open the door to a wider group of users. However, when it comes to information extraction – a classic task in natural language processing – most task-specific systems cannot align well with long-tail ad hoc extraction use cases for non-expert users. To address this, we propose a novel paradigm, termed On-Demand Information Extraction, to fulfill the personalized demands of real-world users. Our task aims to follow the instructions to extract the desired content from the associated text and present it in a structured tabular format. The table headers can either be user-specified or inferred contextually by the model. To facilitate research in this emerging area, we present a benchmark named INSTRUCTIE, inclusive of both automatically generated training data, as well as the human-annotated test set. Building on INSTRUCTIE, we further develop an **O**n-**D**emand **I**nformation **E**xtractor, ODIE. Comprehensive evaluations on our benchmark reveal that ODIE substantially outperforms the existing open-source models of similar size. Our code and dataset are released on https://github.com/yzjiao/On-Demand-IE.

## 1 Introduction

Information extraction has conventionally been divided into a set of well-defined sub-tasks, including named entity recognition (Sang and Meulder, 2003; Levow, 2006; Weischedel et al., 2013), relation extraction (Zhang et al., 2017; Han et al., 2018; Yao et al., 2019), event extraction (Deng et al., 2020; Wang et al., 2020; Zhan et al., 2023) and so on. These components serve as the basis for building complex systems such as virtual assistants and news monitoring systems. However, an average user might also have information extraction needs (as shown in Figure 1) that do not align well with any of the well-defined tasks: the information elements (`shape` and `taste`) might not

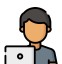
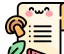

**Instruction**: I personally prefer eating fruits. Extract some key features of the fruits.

**Text**:
Strawberries are a popular fruit known for their vibrant red color and sweet, juicy flavor. …
One of the most widely consumed fruits, apples come in various colors, including red ...
Bananas are elongated, slightly curved fruits that have a thick, protective peel and soft, sweet flesh …

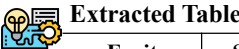

**Extracted Table**:

| Fruit | Shape | Taste | Nutrients |
|---|---|---|---|
| Strawberries | Heart-shaped | Sweet, juicy | Vitamin C, antioxidants |
| Apples | Round | Crisp, sweet | Dietary fiber, vitamin C |
| Bananas | Elongated, curved | Soft, sweet | Potassium, vitamin B6 |

Figure 1: Example of on-demand information extraction. The header of the extracted table can be specified by the user, or inferred by the model.

be covered by existing ontologies and the format might not match any existing task (relation extraction usually involves only two entities and event extraction requires a trigger). Therefore, there is an unfulfilled demand for a more flexible paradigm to extract structured information.

For addressing the shortcomings of conventional information extraction systems for average users, the burgeoning field of Large Language Models (LLMs) offers a promising direction. The latest generation of LLMs is typically subjected to an essential step known as instruction tuning to cater to explicit user commands (Ouyang et al., 2022; OpenAI, 2023). In this stage, models are trained using datasets with specific instructions and the expected responses, which improves LLMs in understanding and reacting to various human queries in natural language. Instruction tuning can be seen as a form of extreme multi-task learning where each input-output pair is a different task, or meta-learning where the model learns to adapt using

the instructions. As a result, these models acquire zero-shot learning ability which emerges as natural interactions with non-expert users.

Instruction-following models unlock access to a broader user base, simultaneously uncovering the diverse requirements in IE scenarios. In light of this, we propose a novel IE paradigm, termed On-Demand Information Extraction, in this paper. Our task is designed to respond to a user's unique instruction and the related text by extracting the sought-after information and presenting it in a user-friendly, structured table format (see Figure 1). It goes beyond the constraints of predefined task settings or ontologies, i.e., the header of the output table can be either personalized by the user or inferred by the model itself from the given text and the instruction. This provides users the flexibility to offer instructions with varying levels of specificity, thereby customizing the output to suit their individual needs. Moreover, On-Demand Information Extraction is not limited to a specific domain, making it a highly scalable and adaptable solution for a wide range of applications.

To benchmark this new task, we construct an instruction-tuning dataset for on-demand information extraction, named INSTRUCTIE. It is comprised of 14,579 training pairs and 150 manually curated test samples, serving as a supplement to the existing open-source instruction-tuning collections. To compile training data and boost the model's capacity to adhere to specific instructions, we employ an automatic generation process via ChatGPT, which generates a range of instructions to ensure wide coverage across diverse domains. We further incorporate a Chain-of-Thought prompting approach (Wei et al., 2022) into our data generation process, enabling us to investigate the effects of elaborating the "thought process" before extracting tables for on-demand IE use cases. More importantly, we propose multi-faceted validation methods to filter out low-quality samples, ensuring that the synthetic data is curated from four perspectives: validity, informativeness, consistency with the instruction, and faithfulness to the text.

Additionally, to evaluate language models empirically, we introduce a set of manually annotated test data from scratch. This dataset showcases a wide range of instructions spanning extremely diverse domains, providing a clear reflection of real user requirements. The background text utilized in this dataset is carefully collected by retrieving from various online sources or in privacy-sensitive cases, generated by the best-performing language model, GPT-4 (OpenAI, 2023). The dataset encompasses different levels of difficulty - part of the instructions further demands comprehensive reasoning and summarization abilities.

On top of INSTRUCTIE, we develop our model, ODIE (**On-D**emand **I**nformation **E**xtractor), which is founded on the LLaMA-7B (Touvron et al., 2023) and LoRA (Hu et al., 2022) techniques. Furthermore, we establish an extensive test bed specifically for this task. Compared to the performance of powerful open-source instruction-following models, ODIE brings substantial improvements in the accuracy of the extracted headers and contents according to automatic metrics and human evaluations. Ablation studies also verify the effectiveness of our proposed filtering method. These findings underscore the potential of INSTRUCTIE and ODIE to facilitate research in on-demand IE scenarios.

## 2 Related Work

### 2.1 Information Extraction

Information extraction is conventionally represented by sub-tasks such as named entity recognition (Lample et al., 2016; Cao et al., 2019; Lin et al., 2019; Huang et al., 2021; Lin et al., 2020; Yu and Ji, 2023), relation extraction (Yu et al., 2017; Sui et al., 2020), event extraction (Ji and Grishman, 2008; Yu et al., 2021; Zhan et al., 2023; Jiao et al., 2023). However, there is a clear gap between the task-specific systems in existing studies and the requirements in open IE scenarios (Li et al., 2023). To cater to a wider group of average users (Ouyang et al., 2023), we propose a novel on-demand information extraction task in this paper. Although our output format mirrors the text-to-table setting (Bao et al., 2018; Wu et al., 2022), our task diverges significantly from previous work in two key aspects: 1) our task hinges on user instructions, enabling personalized extraction, and 2) the headers of the table are not confined to pre-defined types. Instead, they can either be defined by the user or independently inferred by the model based on the context.

### 2.2 Instruction Tuning

Instruction tuning provides a promising solution for finetuning LLMs to better understand and respond to human requests that are expressed in natural language (Ouyang et al., 2022; Sanh et al., 2022). The success of instruction tuning relies

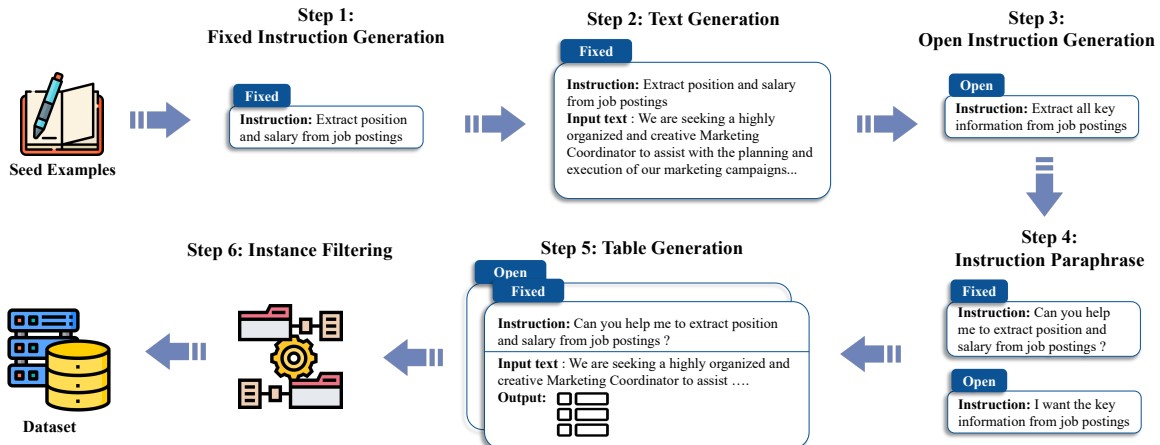

Figure 2: Overall framework of training data generation.

on diverse and representative instruction datasets, which help prepare language models for potential downstream usage (Wang et al., 2023). Existing instruction-tuning datasets are often collected via crowdsourcing (Mishra et al., 2022; Wang et al., 2022b; Databricks, 2023) or via distillation from LLMs (Wang et al., 2022a; Honovich et al., 2022; Taori et al., 2023; Peng et al., 2023). As outlined in Wang et al. (2023), the current landscape of open-source datasets for instruction tuning is predominantly composed of open-ended conversations, reasoning, and coding tasks. Consequently, they fail to adequately prepare the model to follow instructions related to information extraction. Hence, it is imperative to develop a benchmark that is representative of instruction-based information extraction in order to advance the exploration in this field.

## 3 INSTRUCTIE Dataset

In this section, we first formulate the on-demand information extraction task. Following this, we delve into the details of automatically generating training data for INSTRUCTIE, as well as the process of human annotations for the test set.

### 3.1 Task Formulation

Formally, given a user instruction $I$ and associated background text $X$, the model $M$ is designed to extract pertinent information and organize it into a structured table $T$. In the table, the first row represents the table header and the rest is the extracted content. To better align with real user needs, note that the input instructions are not required to explicitly specify the header to be extracted. With this criterion, instructions are categorized into two types: *fixed instruction* and *open instruction*. For *fixed instruction*, the desired headers for the output table are clearly defined, and the model's task is to pull out the relevant content from the text. In contrast, *open instruction* presents a more demanding scenario where the model needs to first infer table headers based on the context, and then extract corresponding information.

### 3.2 Automatic Generation of Training Data

To enhance the instruction-following capability of LLMs in on-demand IE, high-quality training data is of paramount importance. As shown in Figure 2, we devise an automatic pipeline via ChatGPT, encompassing six steps as follows. All the used prompts are listed in Appendix .

**Fixed Instruction Generation.** As instruction is the determining factor in the on-demand use case, we start with fixed instruction collection. Concretely, we provide five manually labeled demonstrations and harness the power of in-context learning (Brown et al., 2020) to guide the model in discerning what constitutes a fixed instruction. In principle, the fixed instruction should specify the header to be extracted, and the type of the text, e.g., "Extract position and salary from job postings". To promote diversity, we require ChatGPT to generate 10 different instructions along with distinct domains for each iteration.

**Background Text Generation.** With a given fixed instruction and domain, the subsequent step involves generating the related background text. We also provide demonstrations and specify in the prompt that the generated text should cover the des-

ignated header and conform to the given text type and domain. For instance, for the fixed instruction in the previous paragraph, ChatGPT is supposed to generate multiple job postings containing different positions and salaries.

**Open Instruction Generation.** After creating the background text, our goal is to produce corresponding open instructions that stand apart from the fixed instruction. An open instruction does not specify the type of header, i.e., "`position`" and "`salary`", but rather employs relatively vague requirements, such as "`key information`", mirroring part of real-world usage. We find that open instructions directly converted from fixed instructions bear strong similarity. Hence, we discard fixed instructions as input, allowing ChatGPT to generate distinct open instructions using only text.

**Instruction Paraphrasing.** To equip the model with the ability to better comprehend user instructions across a diversity of styles, we incorporate an additional paraphrasing step. We delineate four styles: comprehensive query, casual interaction, direct command, and professional request. Each instruction is randomly assigned a style, and Chat-GPT is tasked with paraphrasing it accordingly. It's also emphasized in the prompt that the paraphrased instruction must retain its key elements, specifically the header and the type of text.

**Table Generation.** Given a pair of instruction and background text, the model takes these as input and extracts the corresponding information into a tabular format. Considering the different categories of instructions, we set distinct expectations for the table headers. For fixed instructions, we anticipate that the model adheres strictly to the instruction, extracting only the headers specifically mentioned. Conversely, for open instructions, we aim for an output inclusive of all relevant details. Therefore, the model is tasked with generating as many columns as possible to ensure comprehensive instruction execution. For each input, we generate two versions of the output: 1) *Direct*, which represents the direct output of the table without any accompanying text; and 2) *CoT*, which denotes the introduction of a Chain-of-Thought method (Wei et al., 2022), allowing the model to articulate its thought process prior to table extraction.

**Verification and Filtering.** To uphold the high quality of generated instances, meticulous verification and filtering methods are crucial. Since the final output of the entire pipeline is the generated table, it serves as a comprehensive reflection of the quality of each preceding step. As a result, we center our attention on the generated table to carefully craft a filtering mechanism across four dimensions.

(1) Validity. This checks the validity of the pipeline output, determining whether it conforms to the tabular format. Any instances that don't meet this format are filtered out.

(2) Informativeness. We necessitate that the generated table comprises an adequate number of rows and columns, without containing excessive empty cells, to ensure it offers sufficient information.

(3) Consistency with instruction. For each fixed instruction, the header extracted by the model should strictly align with the provided instruction. To check for consistency with instruction, we formulate it as a Natural Language Inference (NLI) problem. Essentially, Given each header $H$, we construct a sentence such as "extract $H$ from the text" as the hypothesis while the instruction serves as the premise. Taking these two as the input, we adopt a neural evaluator (Zhong et al., 2022) to calculate the factual consistency score. If the average consistency score is below a predefined threshold, the table will be dropped.

(4) Faithfulness to text. For every extracted table, each cell's content should correspond faithfully to the provided background text. We still take this as an NLI task, i.e., given a cell $C$, we utilize its header $H$ to build a hypothesis such as "The $H$ is $C$". Meanwhile, the background text is regarded as the premise. The same neural evaluator calculates the scores, and a threshold is established to select instances that demonstrate high faithfulness.

### 3.3 Manual Annotation for Test Data

We create test set by hand to evaluate how well LLMs perform in the on-demand information extraction task. In the annotation process, GPT-4 (OpenAI, 2023) is employed to assist human annotators, ensuring a wide-ranging collection. The specific process and details are as follows.

**Candidate Domain Creation.** To achieve diversity in test data, we incorporate the vast knowledge base of GPT-4 to enable human annotators to think beyond conventional domains and consider more specialized fields. Specifically, we prompt GPT-4 to generate 1,000 instructions across diverse domains as a candidate pool. Then we hand-pick

150 samples that are not only representative but also align with the genuine needs of different user groups Notably, these generated instructions are not part of the test set but are used as cues to kick off the data-gathering phase.

**Background Text Collection.** Based on candidate instructions, the human annotators proceed to collect the relevant text. The principle is to retrieve real text from the web that meets the requirements. However, for several candidate instructions involving privacy issues (e.g., medical records), we use GPT-4 to generate synthetic data instead. The length of the text is specified to be between 100 and 1,000 words and should contain structured information suitable for extraction.

**Instruction Annotation.** Given the collected text, annotators are required to write corresponding instructions. Standard instructions should be in line with the user input, and diverse in description style. The information required for extraction should be found in the original text. Each instruction is limited to 200 words, and the annotator can determine whether it is an open or fixed instruction before annotating based on the content of the text. In addition, the need for appropriate reasoning or summarization abilities when extracting is permitted, if it is consistent with real-world usage (Concrete examples in Appendix E).

**Table Annotation.** To improve the efficiency of table annotation, we utilize GPT-4 to generate three tables by setting temperature to 1.0 to serve as references for annotators for each input. Annotators can modify or integrate these reference tables into the final result. The content presented in the table should be exhaustive, aligned with the user instruction, and accurately reflect the given background text. For information that is not specified in the original text, "N/A" is uniformly used in the table.

**More Annotation Details.** For the entire process, we invite 3 graduates and 1 undergraduate student with research experience in the field of IE for data annotation. Each instance is annotated by an annotator and then reviewed by a reviewer. The annotator and reviewer discuss together and make necessary modifications until they reach an agreement. After finalizing the annotation, all annotators are gathered to categorize each instance into three difficulty levels, including easy, medium, and hard. We take the following criteria for this

| Data | Train | | | Test |
|---|---|---|---|---|
| | Direct | CoT | Overall | |
| # Instruction | 7,483 | 7,096 | 14,579 | 150 |
| - # Open Instruction | 3,773 | 3,676 | 7,449 | 36 |
| - # Fixed Instruction | 3,710 | 3,420 | 7,130 | 114 |
| # Text | 4,507 | 4,380 | 4,751 | 150 |
| - # Retrieved Text | 0 | 0 | 0 | 119 |
| - # Generated Text | 4,507 | 4,380 | 4,751 | 31 |
| # Domain | 82 | 82 | 83 | 61 |
| # Ave. Instr. Len. | 20.4 | 20.3 | 20.4 | 26.8 |
| # Ave. Text Len. | 281.2 | 277.8 | 279.6 | 310.8 |
| # Ave. Table Cell | 20.8 | 23.2 | 22.0 | 17.1 |
| # Ave. Table Row | 3.6 | 3.7 | 3.6 | 4.7 |
| # Ave. Table Column | 6.2 | 6.7 | 6.5 | 4.1 |
| # Easy Level | - | - | - | 56 |
| # Medium Level | - | - | - | 55 |
| # Hard Level | - | - | - | 38 |

Table 1: Statistics of our dataset, InstructIE. # denotes the number. Ave. denotes the average value. Len. denotes the length in words.

categorization: typically, easy-level instances have fixed instructions and small- or moderately-sized groundtruth tables. Those labeled as medium often include the fixed instructions to extract large-sized and complex tables or a small portion of open instructions. Instances classified as hard generally necessitate the reasoning or summarization abilities and incorporate most of the open instructions.

Regarding the final data format, beyond the instruction, text, and table, we also provide the subsequent human-annotated information:

(1) "Domain" represents the area or field that the data instance pertains to.

(2) "Category" denotes whether the instruction is classified as open or fixed.

(3) "Source type" characterizes the source of background text, classifying it as either "retrieve" - sourced from a real-world text, or "generate" - produced by GPT-4.

(4) "Difficulty level" indicates the complexity of the extraction for the current sample from the human perspective, and is divided into three levels: "easy", "medium", and "hard".

### 3.4 Dataset Analysis

INSTRUCTIE dataset consists of 14,579 samples for training and 150 for testing, with the statistics displayed in Table 1. Notably, the scale of our test set aligns with existing instruction-tuning benchmarks. For instance, AlpacaFarm (Dubois et al., 2023), a collection of existing instruction-tuning test sets, encompasses 90 instructions from the Vicuna evaluation (Chiang et al., 2023), 129 from

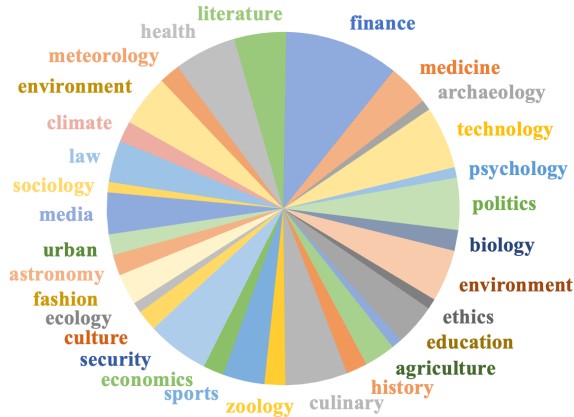

Figure 3: Representative domains of testing data.

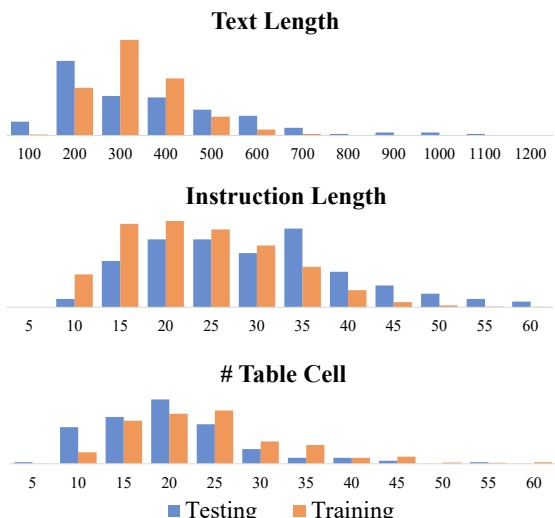

Figure 4: Distribution of training and testing data.

Anthropic (Bai et al., 2022), 156 from the Koala evaluation (Geng et al., 2023), 188 from Open Assistant (Köpf et al., 2023), and 252 from the evaluation of self-instruct (Wang et al., 2022a). However, these benchmarks primarily focus on open-ended dialogues, reasoning, and coding tasks, and hence, INSTRUCTIE can supplement these sources by focusing on the aspect of information extraction.

In particular, our training set comprises 7,483 direct instances and 7,096 CoT instances. The dataset, entirely composed of model-generated instructions and texts, extensively spans 84 domains (listed in Appendix A). Our analysis reveals that the tables have an average of 22 cells after filtering. Interestingly, on comparing the direct and CoT methods, we observe that the tables produced via CoT approach contain an average of 3 more cells than those generated directly.

As for the testing data, the instructions are categorized into two types - 36 open instructions and 114 fixed instructions. We retrieve 119 real texts, alongside 31 generated texts in areas that are sensitive to privacy. As illustrated in Figure 3, 150 testing cases span a broad range of 61 domains. The manually annotated tables contain 17 cells on average. Figure 4 compares the distribution in text length, instruction length, and the average number of cells in tables between the training and testing data. Moreover, the test set of INSTRUCTIE has exhibits a well-balanced distribution across three levels of difficulty, comprising 56, 55, and 39 instances respectively.

## 4 Experiment

In this section, we first introduce our instruction-tuned model, thendescribe the experimental setup

and finally present the detailed results.

### 4.1 Model Training - ODIE

To establish a model with instruction-following capability on IE, we finetune LLaMA-7B (Touvron et al., 2023) with LoRA (Hu et al., 2022), a parameter-efficient fine-tuning technique, on the training set of INSTRUCTIE to obtain ODIE. We format the datasets to follow a chatbot-style schema to allow interactions between the user and the language model (a.k.a. "assistant") into one input sequence. Concretely, each instance begins with the "<|system|>" token, accompanied by distinct system prompts for both Direct and CoT methods[1]. This is succeeded by the "<|user|>" token and the input instruction, after which the model's response follows the "<|assistant|>" token. During training, we compute the cross entropy loss only on tokens after <|assistant|>.

For a fair comparison, we adopt identical training paradigms, hyperparameters, and backbone models for training ODIE and all other baseline models. The only exception is the number of epochs, as the optimal number of training steps required to achieve peak performance varies depending on the size of the training data. More details about training can be found in Appendix D.

---

[1] System prompt for the CoT method is "You are a helpful assistant. Follow the user instruction to output a paragraph as the explanation and extract information from the given text into a concise markdown table." For the direct method, "output a paragraph as the explanation" is removed.

## 4.2 Evaluation Metrics

Since the on-demand information extraction task is formulated as the text-to-table generation, we divide the evaluation into two parts: evaluating the table headers and table contents. Table headers reflect how well the model understands user instructions. We adopt a soft matching strategy (Jiao et al., 2022) by using SentenceBERT (Reimers and Gurevych, 2019) to calculate the cosine similarity as the semantic similarity score. The table contents reflect the quality of extraction. We use the ROUGE-L $F_1$ score (Lin, 2004) to evaluate the generated output. We also conduct human evaluations to provide a comprehensive assessment.

## 4.3 Evaluated Models

We compare three categories of models: open-source instruction-following models, proprietary models, and models trained on INSTRUCTIE.

**Public Instruction-Following Models.** ALPACA and TÜLU are state-of-the-art instruction-following models, with instruction tuning on 52K and 512K data collections (Databricks, 2023; Longpre et al., 2023; Köpf et al., 2023; Peng et al., 2023; Chaudhary, 2023), respectively. TÜLU, in particular, stands out as the best-performing model at the 7B size (Wang et al., 2023). We acquire these two baseline models by training on publicly accessible datasets using LoRA technique.

**Proprietary Models.** We select the two best LLMs, CHATGPT and GPT-4 (OpenAI, 2023) for comparison. Since their training data is not disclosed and the model size is much larger than ours, we mainly use their performance as the reference.

**Our Models.** Depending on whether the CoT method is added during training, we include two versions of our model ODIE-DIRECT and ODIE-CoT. Additionally, we report two variants obtained by removing the filtering process of training data.

## 4.4 Table Header Evaluation

We report the soft matching scores for both fixed and open instructions in Table 2. Our model showcases highly competitive performance: ODIE-DIRECT outperforms the best open-source baseline by 4%, and nearly matches the performance of GPT models despite the significant difference in model sizes. This outcome verifies that our models effectively develop the instruction-following capability for on-demand IE. Notably, ODIE-CoT

| Models | Category | | Overall |
| | Fixed | Open | |
|---|---|---|---|
| **Open-Source Models** | | | |
| ALPACA | 65.89 | 45.69 | 59.80 |
| TÜLU | 77.78 | 49.26 | 69.44 |
| **Our Models** | | | |
| ODIE-DIRECT | **83.59** | 51.67 | **73.82** |
| - Filtering | 83.54 | 51.37 | 73.61 |
| ODIE-CoT | 72.32 | **54.17** | 66.81 |
| - Filtering | 68.97 | 53.01 | 64.11 |
| **Proprietary Models** | | | |
| CHATGPT | 81.69 | 57.86 | 74.49 |
| GPT-4 | 82.06 | 57.78 | 74.47 |

Table 2: Experimental results for header evaluation. The metric is $F_1$ (%) of the soft matching score.

demonstrates an even better performance in the context of open instructions, surpassing ODIE-DIRECT by 2.5% while it shows a weaker performance on fixed instructions. Based on careful observation, we speculate that this is because the CoT stimulates the model to think more dynamically and adaptable, which is particularly beneficial for open instructions that can have different formulations and contexts. However, when it comes to fixed instructions, CoT might grant the model excessive flexibility, exacerbating the issue of violating the instruction and generating additional headers (Figure 8 in the Appendix shows case study on comparing ODIE-DIRECT and ODIE-CoT).

## 4.5 Table Content Evaluation

Table 3 presents evaluation results for table content analysis, showing that our ODIE models outperform open-source baselines substantially. Notably, ODIE-DIRECT performs 5.4% better than the best baseline model, TÜLU. Among our ODIE models, those with multi-facet filtering bring at least 2.6% higher performance than versions without filtering, validating the effectiveness of our proposed method. However, despite these improvements, all open-source models, including ours, are not on par with state-of-the-art proprietary models in the table content evaluation.

We further conduct an in-depth analysis examining various models across three aspects: instruction categories, text sources, and difficulty levels.

**Instruction category**: All models perform considerably better on fixed instructions as opposed to open instructions. This is expected since open instructions demand more sophisticated text analysis and extraction abilities.

**Text source**: Models find the retrieved context

| Models | Difficulty | | | Category | | Source | | Overall | # Data |
|---|---|---|---|---|---|---|---|---|---|
| | Easy | Medium | Hard | Fixed | Open | Generate | Retrieve | | |
| **Open-Source Models** | | | | | | | | | |
| ALPACA | 26.27 | 20.08 | 22.72 | 25.29 | 16.07 | 30.37 | 21.18 | 23.08 | 52K |
| TÜLU | 43.69 | 39.15 | 38.68 | 42.55 | 34.94 | 45.08 | 39.59 | 40.72 | 512K |
| **Our Models** | | | | | | | | | |
| ODIE-DIRECT | **48.01** | **45.38** | **43.71** | **47.19** | **41.92** | **49.49** | **45.00** | **45.93** | 7.5K |
| - Filtering | 46.31 | 39.85 | 42.44 | 44.42 | 38.23 | 46.71 | 41.95 | 42.93 | 7.5K |
| ODIE-CoT | 44.47 | 39.99 | 41.73 | 43.02 | 39.25 | 49.01 | 40.32 | 42.12 | 7.1K |
| - Filtering | 41.26 | 36.59 | 41.20 | 40.08 | 38.93 | 45.92 | 37.87 | 39.53 | 7.1K |
| **Proprietary Models** | | | | | | | | | |
| CHATGPT | 52.45 | 50.56 | 51.07 | 53.21 | 45.66 | 55.97 | 50.21 | 51.40 | - |
| GPT-4 | 60.78 | 55.76 | 61.24 | 61.51 | 51.29 | 65.89 | 57.28 | 59.06 | - |

Table 3: Experimental results for table content evaluation. The metric is $F_1$ score of ROUGE-L (%).

more challenging, as the generated text usually follows a more structured format, which makes it easier to identify the key information.

**Difficulty level**: Most models tend to perform best on easy instances, yet intriguingly, part of the models exhibit better performance in hard-level samples compared to medium-level ones. After manual analysis, we attribute this to the fact that medium-level instances usually require extracting complex and large-size tables, where models often fail to capture all essential elements in the text, resulting in lower scores. Conversely, hard-level samples demand reasoning or summarization ability for specific rows, columns, or cells. For instance, when extracting the information from recipes, models may struggle with calculating the cooking time of all the steps, but still produce a reasonable table. It would not cause significant impacts on the score. This discrepancy between human judgment and model performance calls for effort on the more fine-grained evaluation metrics.

## 4.6 Human Evaluation

For human evaluation, we ask three annotators of the instructions to evaluate model outputs. The evaluators are asked to rate the output of four instruction-following models based on whether the extracted information is accurate. In line with automatic evaluations, we evaluate the header and content of the table separately. The details and the annotation platform are presented in Appendix C.

For the table headers, we design a three-level rating system for the models' outputs: (1) Rating-A: correct, (2) Rating-B: partly correct, or (3) Rating-C: completely wrong. Figure 5(a) illustrates the performance of four models, with ODIE showing comparable results to GPT-4, both receiving only 9 votes in Rating C. ODIE also outperforms TÜLU

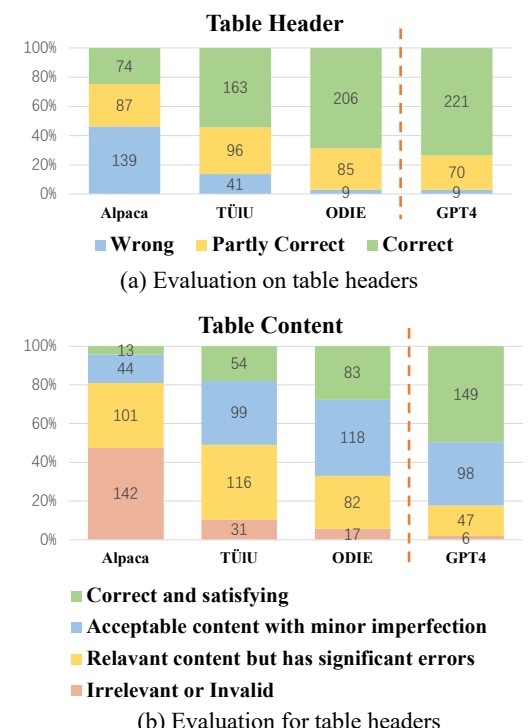

(a) Evaluation on table headers

(b) Evaluation for table headers

Figure 5: Human evaluations on headers and contents.

in terms of correct and partly-correct scores. Conversely, ALPACA struggles to follow instructions, frequently producing invalid results.

For the table contents, we set up four-level rating standards, including (1) Rating-A: valid and satisfying, (2) Rating-B: acceptable but has minor errors or imperfections, (3) Rating-C: relevant to the instruction with significant errors, and (4) Rating-D: irrelevant or completely invalid. As shown in Figure 5(b), GPT-4's much larger model size allows it to excel beyond the other models, while ODIE continues to display clear benefits over ALPACA and TÜLU, particularly in Rating A and B.

| Metrics | Pearson | Spearman | Kendall |
|---|---|---|---|
| [Header] Exact Match. | 0.640 | 0.609 | 0.494 |
| [Header] Semantic Sim. | 0.817 | 0.769 | 0.637 |
| [Content] Exact Match. | 0.338 | 0.375 | 0.272 |
| [Content] Semantic Sim. | 0.764 | 0.705 | 0.558 |
| [Content] RougeL | 0.713 | 0.704 | 0.554 |

Table 4: Correlation Analysis. To further study different metrics on our proposed task, we analyze the correlation between these automatic metrics and human evaluations. Here Exact Match. means exact matching and Semantic Sim. means semantic similarity. The analysis results includes the Pearson, Spearman, and Kendall coefficients.

### 4.7 Evaluation Metric Analysis

To make our evaluation comprehensive, we study two automatic evaluation metrics on table headers (Exact Match and Semantic Similarity) and three ones on table content (Exact Match, Semantic Similarity, and ROUGEL). The experiments results can be found in Tables 7-10 in the appendix, which show a similar trend. On these different metrics, our methods can still obviously outperform the open-source baselines while being comparable with GPT4 on table header generation.

To discover the suitable automatic evaluation metric for our proposed task, we further analyze the correlation between these automatic metrics and human evaluations. Table 4 lists the analysis results, including the Pearson, Spearman, and Kendall coefficients. The results indicate that, for table header, two studied metrics are highly correlated with human evaluation. But for table content, RougeL and semantic similarity can indeed be more reliable metrics for this task compared with exact matching due to significantly high correlation. In this paper, we show RougeL for its wide use in prior instruction-tuning work (Wang et al., 2023).

### 5 Conclusion

This paper introduces a new task, On-Demand Information Extraction to fulfill users' personalized needs by extracting content based on instructions and organizing it in a table with user-specified or model-inferred headers. To benchmark this new task, we construct a comprehensive dataset INSTRUCTIE including synthesized training data and human-annotated test data. Our developed model ODIE outperforms existing open-source models in extensive experiments.

## Limitations

While this paper contributes to the research field by introducing the On-Demand Information Extraction task and constructing the INSTRUCTIE dataset, it still has the following limitations:

1. **Model Size and Data Constraints**: The experiments presented in this paper primarily focus on the utilization of the 7B model. Due to the limited computing resources, an exploration into the impact of varying model sizes and the potential benefits of using larger datasets could not be conducted. It remains an open question how scalability in terms of model size and data volume would affect the performance and efficiency of the On-Demand Information Extraction task.

2. **Combination of Direct and CoT Methods**: In our experimental analysis, the Direct and CoT method are discussed and evaluated separately in the On-Demand Information Extraction task. However, the potential synergistic effects of combining both methods have not been investigated. It could possibly yield insights into different dimensions of the task or further improve the model performance.

3. **Evaluation Metrics**: The evaluation metrics used in the current experiments are primarily detecting the overall similarity between the model outputs and the groundtruth. However, given the flexible nature of table structures, it is imperative to have evaluation metrics that can assess the accuracy and quality of table construction and information organization in a more fine-grained manner. Developing more effective and precise evaluation metrics is necessary to robustly evaluate different aspects of our On-Demand Information Extraction task.

4. **Contextual Inference and Complex Instructions**: As highlighted in the conclusion, the current model has room for improvement in contextually inferring table structures and processing complex instructions. This limitation can affect the utility and user experience, particularly for users who are not domain experts and may not know how to frame their queries optimally. Enhancing the model's capabilities in these areas is essential for ensuring that the On-Demand IE task is accessible and friendly for a wide range of users.

In light of these limitations, future work should focus on exploring the impact of model size and data scale, investigating the combination of different data types, developing more nuanced evaluation metrics, and improving the model's ability to infer context and handle complex instructions. These efforts are crucial in advancing the On-Demand Information Extraction systems and making them more accurate and widely applicable.

## Ethics Statement

In conducting the research presented in this paper, we adhere to ethical standards and principles to ensure the integrity and validity of our work. Specifically, the dataset INSTRUCTIE constructed for this research is developed with utmost care to ensure that no personal or sensitive information is included. The human-annotated test data is collected and used in compliance with relevant ethical guidelines. Additionally, during the data collection process, we verify the licenses of the data source websites to ensure that our use of the data sticks to the terms and conditions stipulated by the data providers. Moreover, during the development of the dataset, we mitigate any biases that may arise, ensuring that the data is representative and does not favor any particular group or perspective.

## Acknowledgements

Research was supported in part by US DARPA KAIROS Program No. FA8750-19-2-1004 and INCAS Program No. HR001121C0165, National Science Foundation IIS-19-56151, IIS-17-41317, and IIS 17-04532, and the Molecule Maker Lab Institute: An AI Research Institutes program supported by NSF under Award No. 2019897, and the Institute for Geospatial Understanding through an Integrative Discovery Environment (I-GUIDE) by NSF under Award No. 2118329. Any opinions, findings, and conclusions or recommendations expressed herein are those of the authors and do not necessarily represent the views, either expressed or implied, of DARPA or the U.S. Government.

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

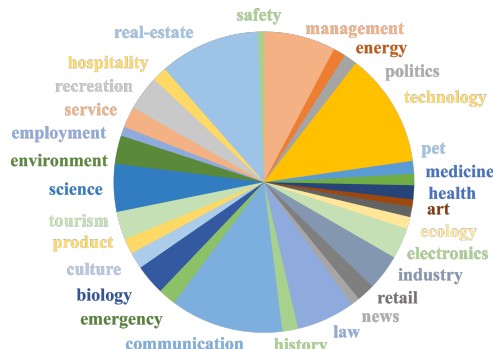

Figure 6: Representative domains of training data.

*Abu Dhabi, United Arab Emirates, December 7-11, 2022*, pages 2023–2038. Association for Computational Linguistics.

## A  Domain of Training Data

In our INSTRUCTIE dataset, all the instructions, and texts in this dataset are model-generated and extensively involve 84 domains (see Figure 6), such as management, ecology, marketing, health, astronomy, meteorology, beauty, design, linguistics, and hospitality.

## B  Prompting Templates for Data Generation

The INSTRUCTIE dataset relies on a number of prompting templates in order to elicit the generation from language models for training data. Here we provide our five templates for 1) fixed instruction generation, 2) background text generation, 3) open instruction generation, 4) instruction paraphrasing, and 5) table generation (Table 5 and Table 6). Note that during the stage of test data annotation, we use the same prompts to generate instruction hints for Candidate Domain Creation, generate texts for Background Text Collection, and generate reference tables for Table Annotation.

## C  Human Evaluation Details

### C.1  Human Evaluation Setup

Here we provide more details for the human evaluation described in the experiment for rating the models' responses to the 150 user-oriented information extraction instructions. Human evaluation in done using a open source annotation tool, doccano[2]. To ensure faithful and reliable evaluation, we asked

---

[2]https://github.com/doccano/doccano

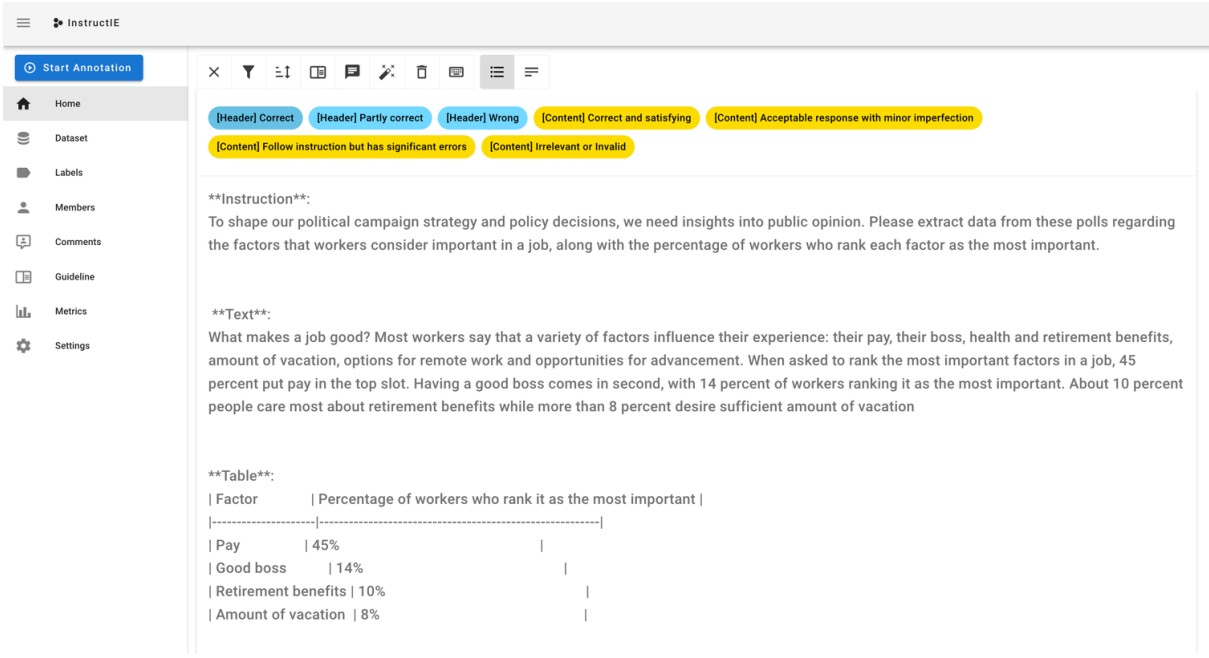

Figure 7: Annotation interface for human evaluation. The predictions from different models present in random order and the model information being anonymized. Our expert evaluators are required to read the instruction and input, refer to the target, and then select the rating for the model's outputs from three options for the table headers and fours options for the table contents

three authors of these instructions (and of this paper) to judge model predictions. These three evaluators coordinated the standards for the rating system before starting annotation and then each of them rated all the instances independently. They were presented with the instruction, the background text, the target output (as a reference), and the model outputs. Model responses are listed in random order, with all the model information anonymized. Figure 7 provides a screenshot of the annotation interface. The reported performance in this paper is based on the results from all evaluators.

### C.2 Human Evaluation Agreement

To measure how reliable our human evaluation is, we calculate the inner-rater agreement among our three evaluators. We first report the Fleiss's kappa value[3], which is commonly used to measure inter-rater agreement for categorical items. When calculating this, we separately handle the table headers and contents. For the table headers, we treat the 3-level rating as a categorical variable, leading to a kappa value of 0.55, which is a moderate agreement according to common practice. Furthermore, we also calculate the agreement of our evaluators on 4-level rating for table contents, 0.49, which

also indicates moderate agreement.

## D Implementation Details

### D.1 Training Data Generation

In this part, we introduce the parameters and details involved in the phase of generating training data. In our pipeline, for the first step, Fixed Instruction Generation, we prompt ChatGPT to generate 10 different fixed instructions in each iteration, doing this a total of 500 times, resulting in 5000 instructions along with their corresponding domains. Each domain here is represented by a single word. In the second step, Background Text Generation, we generate a piece of text corresponding to each instruction, resulting in 5000 pieces of text. In the third step, Open Instruction Generation, we generate an open instruction for each piece of text. By combining the two types of instructions, we obtain a total of 10,000 training pairs. In the fourth step, Instruction Paraphrasing, we sample ten instructions each time, regardless of whether they are fixed or open, and pass them together to ChatGPT for paraphrasing. ChatGPT is required to output the same number of instructions. If the number of outputs from GPT does not match, the results of that round will be discarded. This means that we query ChatGPT more than 1000 times in total.

---

[3]https://en.wikipedia.org/wiki/Fleiss%27_kappa

In the fifth step, Table Generation, we generate a table for each of the 10,000 training pairs. After the above steps, we obtain 10,000 raw direct and CoT instances. In the sixth step, Verification and Filtering, considering Validity, 438 direct data instances are filtered out. To ensure Informativeness, we require the sum of the number of rows and columns in the table to be greater than 3, and the number of columns to be more than 1. Additionally, the number of "N/A" in the table should be less than 4. Tables that do not meet these requirements, totaling 653, are discarded. To ensure Consistency and Faithfulness, we set a threshold equal to 0.5. Only the tables with average scores greater than the threshold are retained. These two steps filtered out 552 and 874 direct data instances respectively. As for CoT, these four filtering strategies remove 961, 250, 696, and 997 instances respectively.

## D.2 Model Training

We use LLaMA-7B (Touvron et al., 2023) as the backbone model and finetune it with the LoRA approach (Hu et al., 2022) for all the models. During training, we configure the batch size to 64 and the maximum learning rate to 3e-4 with a 0.03 warmup ratio. For all the experiments, the LoRA $r$ is set to 16, and we apply a dropout rate of 0.05. We keep these hyperparameters the same for a fair comparison. Therefore, the only differences among ALPACA, TÜLU, and ODIE lie in the instruction-tuning data utilized for training and the number of training epochs. Considering the different amounts of training data, we train ALPACA 5 epochs (52K data), TÜLU 2 epochs (512K data), and ODIE 20 epochs (7K data) respectively to achieve the best performance on INSTRUCTIE.

During the inference process, we also adhere to the same set of parameters: a temperature of 0.1, top_p of 0.75, top_k of 40, 4 beams, and a maximum generation length of 2,048.

## E Case Study

Our case study is conducted from two perspectives, the type of the model and the difficulty level of the dataset. Figure 8 shows the output of our two models, ODIE-DIRECT and ODIE-COT, for the same instruction. The instruction is fixed, so we expect the model to follow the instruction and extract the information specified by the instruction, which are patterns and indicators. According to the observation of model outputs, we can find that

ODIE-DIRECT can extract a decent table while ODIE-COT violates the instruction and generates five irrelevant headers, including the type of incident, affected company, data, location, and other details. We speculate that this is because the CoT stimulates the model to think more dynamically and adaptably, which is particularly beneficial for open instructions that can have different formulations and contexts.

Figure 9, 10, and 11 respectively show three examples of different difficulty levels. The first example is at the easy level, which involves fixed instruction and small-sized groundtruth tables. According to the model outputs, we can see that most of the four models can extract the mostly-correct information. The second medium example is a fixed instruction to extract a relatively large-sized and complex table. In this case, we find that the two open-source baselines miss some rows or columns. In contrast, our model can extract more useful information. But even for GPT4, there is a pattern in its output that does not match the groundtruth. The third example is the hard case, whose instruction requires commonsense reasoning, which requires the model to understand the color of food and find out the orange ones. According to the experimental results, all models fail to exclude non-orange Strawberry Applause. But in general, they were able to extract the recipe's name, the ingredients, and the approximate cooking time.

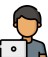 **Instruction**: Please conduct an analysis of the logs and security incident reports. Our objective is to identify patterns and indicators that suggest the presence of cyber threats..

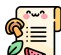 **Text**:

Rapid7 analyzed a sample webshell payload associated with successful exploitation. The webshell code would first determine if the inbound request contained a header named X-siLock-Comment, and would return a 404 'Not Found' error if the header was not populated with a specific password-like value. As of June 1, 2023, all instances of Rapid7-observed MOVEit Transfer exploitation involve the presence of the file human2.aspx in the wwwroot folder of the MOVEit install directory (human.aspx is the native aspx file used by MOVEit for the web interface).Successful exploitation allows attackers to download and steal sensitive information such as list of stored files, usernames who uploaded the files, file paths, configured Azure Blob Storage accounts, data from Azure Blob Storage containers, data from servers, and so on. The attackers can also insert and delete a new random named MOVEit Transfer user with the login name 'Health Check Service' and create new MySQL sessions. Progress Software is advising MOVEit customers to check for indicators of unauthorized access over "at least the past 30 days", as well as other remediation recommendations.

---

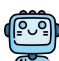 **Model Output of ODIE-Direct**:

| Pattern | Indicator |
|---------|-----------|
| Webshell payload | Determines if X-siLock-Comment header is populated with specific password-like value before returning 404 'Not Found' error |
| MOVEit Transfer exploitation | Presence of human2.aspx file in wwwroot folder of MOVEit install directory, allows attackers to download and steal sensitive information, insert and delete new random named MOVEit Transfer user, and create new MySQL sessions |
| Advice from Progress Software | Check for indicators of unauthorized access over "at least the past 30 days", as well as other remediation recommendations |

---

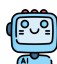 **Model Output of CoT**:

To identify patterns and indicators of cyber threats from the logs and security incident reports, we need to extract information related to cyber attacks, data breaches, malware infections, and other types of security incidents. We will structure this information in a tabular format that clearly shows the type of incident, the affected company or organization, the date and time of the incident, the location of the incident, and any other relevant details that may suggest the presence of a cyber threat.

| Type of Incident | Affected Company/Organization | Date and Time | Location | Other Details |
|------------------|-------------------------------|---------------|----------|---------------|
| Cyber Attack | Progress Software | N/A | N/A | Successful exploitation allows attackers to download and steal sensitive information such as list of stored files, usernames who uploaded the files, file paths, configured Azure Blob Storage accounts, data from Azure Blob Storage containers, data from servers, and so on. The attackers can also insert and delete a new random named MOVEit Transfer user with the login name 'Health Check Service' and create new MySQL sessions. |

Figure 8: Case study on comparing ODIE-DIRECT and ODIE-COT. For the fixed setting, ODIE-DIRECT is better at following instructions, which aims to extract the pattern and indicator. But ODIE-COT is facing the exacerbating issue of violating the instruction and generating five irrelevant headers, including the type of incident, affected company, data, location, and other details.

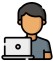 **Instruction**: Retrieve product information such as product name, target audience, key feature, material, and price from e-commerce websites. We need this data organized for categorization and comparison purposes.

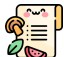 **Text**:

Gildan Cotton Crew Neck T-Shirts are anything but basic. Crafted with soft, breathable cotton and moisture wicking technology to keep you cool and comfortable no matter what's on the agenda. Perfect for layering or lounging - you can thank us later. Price:$18.99.

C4 Energy Drink now available in a delicious and sugar-free carbonated form. This drink features 200mg of caffeine, CarnoSyn beta-alanine and citrulline malate in every serving. Carbonated C4 Original On the Go packs the legendary energy, endurance and pumps that you know and love from C4. Available in a 18 pack with two flavors, Frozen bombsicle and Strawberry Watermelon.Price:$33.59

---

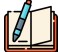 **Groundtruth**:

| Product Name | Target Audience | Key Feature | Material | Price |
|---|---|---|---|---|
| Gildan Cotton Crew Neck T-Shirts | N/A | Moisture wicking technology | Cotton | $18.99 |
| C4 Energy Drink | N/A | 200mg of caffeine, beta-alanine | Carbonated drinks | $33.59 |

---

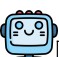 **Model Output of Alpaca**:

| Product Name | Target Audience | Key Feature | Material | Price |
|---|---|---|---|---|
| Gildan Cotton Crew Neck T-Shirts | Everyone | Soft, breathable cotton | Moisture wicking | $18.99 |
| C4 Energy Drink | Everyone | Sugar-free | Carbonated | $33.59 |
| C4 Energy Drink | Everyone | Sugar-free | Carbonated | $33.59 |

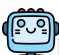 **Model Output of TÜlu**:

| Product Name | Target Audience | Key Feature | Material | Price |
|---|---|---|---|---|
| Gildan Cotton Crew Neck T-Shirts | Clothing | Soft, breathable cotton | Cotton | $18.99 |
| C4 Energy Drink | Energy drink | 200mg caffeine, CarnoSyn beta-alanine, citrulline malate | Carbonated drinks | $33.59 |

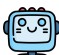 **Model Output of ODIE**:

| Product Name | Target Audience | Key Feature | Material | Price |
|---|---|---|---|---|
| Gildan Cotton Crew Neck T-Shirts | Anyone looking for comfortable layering or lounging | Soft, breathable cotton, moisture wicking technology | Cotton | $18.99 |
| C4 Energy Drink | Anyone looking for a sugar-free carbonated energy drink | 200mg of caffeine, CarnoSyn beta-alanine and citrulline malate | Carbonated water, sugar-free carbonated water, maltodextrin, citric acid, natural and artificial flavors | $33.59 |

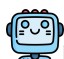 **Model Output of GPT4**:

| Product Name | Target Audience | Key Feature | Material | Price |
|---|---|---|---|---|
| Gildan Cotton Crew Neck T-Shirts | All | Moisture wicking technology | Cotton | $18.99 |
| C4 Energy Drink | All | 200mg of caffeine, beta-alanine | Carbonated drinks | $33.59 |

Figure 9: Case study on easy-level instruction.

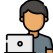 **Instruction**: Please conduct an analysis of the logs and security incident reports. Our objective is to identify patterns and indicators that suggest the presence of cyber threats..

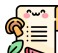 **Text**:

Rapid7 analyzed a sample webshell payload associated with successful exploitation. The webshell code would first determine if the inbound request contained a header named X-siLock-Comment, and would return a 404 'Not Found' error if the header was not populated with a specific password-like value. As of June 1, 2023, all instances of Rapid7-observed MOVEit Transfer exploitation involve the presence of the file human2.aspx in the wwwroot folder of the MOVEit install directory (human.aspx is the native aspx file used by MOVEit for the web interface).Successful exploitation allows attackers to download and steal sensitive information such as list of stored files, usernames who uploaded the files, file paths, configured Azure Blob Storage accounts, data from Azure Blob Storage containers, data from servers, and so on. The attackers can also insert and delete a new random named MOVEit Transfer user with the login name 'Health Check Service' and create new MySQL sessions. Progress Software is advising MOVEit customers to check for indicators of unauthorized access over "at least the past 30 days", as well as other remediation recommendations.

---

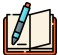 **Groundtruth**:

| Pattern | Indicator |
|---|---|
| Webshell payload | Presence of X-siLock-Comment header with password-like value |
| MOVEit Transfer exploitation | File human2.aspx in wwwroot folder of MOVEit install directory |
| Stealing sensitive information | List of stored files, usernames, file paths, etc. |
| Unauthorized access | Health Check Service as a random named MOVEit Transfer user |
| New MySQL sessions | Creation of new MySQL sessions by attackers |

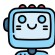 **Model Output of Alpaca**:

| Threat Patterns |
|---|
| Presence of X-siLock-Comment header\n- 404 Not Found error |
| File human2.aspx in wwwroot folder\n- Successful exploitation |
| Attackers can download and steal sensitive information |
| Attackers can insert and delete a new random MOVEit Transfer user\n- Advisory to check for indicators of unauthorized access over "at least the past 30 days |

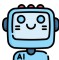 **Model Output of TÜLU**:

| Cyber Threat | Indicator |
|---|---|
| Webshell Payload | X-siLock-Comment header with specific password-like value |
| MOVEit Transfer Exploitation | Presence of human2.aspx file in wwwroot folder of MOVEit install directory |

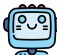 **Model Output of ODIE**:

| Pattern | Indicator |
|---|---|
| Webshell payload | Determines if X-siLock-Comment header is populated with specific password-like value before returning 404 'Not Found' error |
| MOVEit Transfer exploitation | Presence of human2.aspx file in wwwroot folder of MOVEit install directory, allows attackers to download and steal sensitive information, insert and delete new random named MOVEit Transfer user, and create new MySQL sessions |
| Advice from Progress Software | Check for indicators of unauthorized access over "at least the past 30 days", as well as other remediation recommendations |

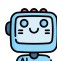 **Model Output of GPT4**:

| Pattern | Indicator |
|---|---|
| Webshell payload | Inbound request containing a header named X-siLock-Comment, returns a 404 'Not Found' error if not populated with a specific password-like value |
| File Presence | Instances of Rapid7-observed MOVEit Transfer exploitation involve the presence of the file human2.aspx in the wwwroot folder of the MOVEit install directory |
| Stolen Information | Successful exploitation allows attackers to download and steal sensitive information such as list of stored files, usernames who uploaded the files, file paths, configured Azure Blob Storage accounts, data from Azure Blob Storage containers, data from servers, etc. |
| Unauthorized access | Attackers can insert and delete a new random named MOVEit Transfer user with the login name 'Health Check Service' and create new MySQL sessions |
| Remediation Recommendations | Progress Software advises MOVEit customers to check for indicators of unauthorized access over at least the past 30 days |

Figure 10: Case study on medium-level instruction.

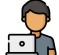 **Instruction**: What are the recipes I can make for my picky toddler who only eat food with orange color? Provide the recipe, ingredients, and cooking time.

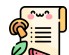 **Text**:

PUMPKIN BREAD: Preheat oven to 325 degrees Fahrenheit (165 degrees Celsius) and grease a 9×5-inch loaf pan. In a large bowl, beat the oil and honey together together with a whisk. Add the eggs and whisk until blended. (If your coconut oil solidifies on contact with cold ingredients, simply let the bowl rest in a warm place for a few minutes, like on top of your stove, or warm it for about 10 seconds in the microwave.) Add the pumpkin purée, milk, pumpkin spice, baking soda, vanilla and salt, and whisk to blend. Lastly, switch to a big spoon and stir in the flour, just until combined. Some lumps are ok! If you're adding any additional mix-ins, gently fold them in now. Pour the batter into your greased loaf pan and sprinkle lightly with cinnamon. If you'd like a pretty swirled effect, run the tip of a knife across the batter in a zig-zag pattern. Bake for 55 to 60 minutes, or until a toothpick inserted into the center comes out clean (typically, if I haven't added any mix-ins, my bread is done at 55 minutes; if I have added mix-ins, it needs closer to 60 minutes). Let the bread cool in the loaf pan for 10 minutes, then carefully transfer the bread to a cooling rack to cool for 20 minutes before slicing.

TURKEY + CARROT ROLL-UP: Place the tortilla on a cutting board. Spread the hummus over the entire surface of the tortilla. On the bottom 2/3 of the tortilla, sprinkle the carrots. Pat down to make them stick in the hummus. Lay the turkey lunch meat over the carrots, but avoid the top 1/3 of the tortilla. Starting with the bottom of the tortilla, with the carrots and turkey, tightly roll the tortilla all the way to the top. The top 1/3 section of hummus will act like glue to help the tortilla stay in place. Cut into 1-inch sections. The whole process needs 10 min.

Strawberry Applause: In a large stockpot, bring the cherries, water, cinnamon, and vanilla to a boil. Reduce to a simmer over medium-low heat, cover, and cook for 15 minutes, stirring occasionally. Add in the strawberries and stir. Cover and then cook for another 15 minutes. Let cool slightly. For a chunky sauce, mash the cherries and strawberries with the back of a wooden spoon. For a smoother sauce, transfer all ingredients into a blender or food processor and puree for 30-60 seconds or until you achieve your desired consistency. I pureed mine for 45 seconds. If pureeing for baby, puree for 1-2 minutes or until very smooth. Transfer to an air-tight container and serve.

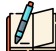 **Groundtruth**:

| Recipe Name | Ingredients | Cooking Time |
|---|---|---|
| PUMPKIN BREAD | oil, honey, pure milk, pumpkin spice, baking soda, vanilla, salt, flour | 80 min |
| TURKEY + CARROT ROLL-UP | Tortilla, hummus, shredded carrots, turkey lunch meat | 10 min |

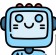 **Model Output of Alpaca**:

| Recipe Name | Ingredients | Cooking Time |
|---|---|---|
| Pumpkin Bread | 1 cup pumpkin purée | 55-60 minutes |
| Turkey + Carrot Roll-Up | Tortilla | 20 min |
| Strawberry Applause | Cherries, water, cinnamon, vanilla | 15 minutes |

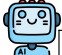 **Model Output of TÜlu**:

| Recipe Name | Ingredients | Cooking Time |
|---|---|---|
| Pumpkin Bread | Oil, honey, eggs, pumpkin purée, milk, pumpkin spice, baking soda, vanilla, salt | 55-60 minutes |
| Turkey + Carrot Roll-Up | Tortilla, hummus, carrots, turkey lunch meat | 10 min |
| Strawberry Applause | Cherries, water, cinnamon, vanilla | 15 minutes |

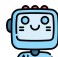 **Model Output of ODIE**:

| Recipe Name | Ingredients | Cooking Time |
|---|---|---|
| Pumpkin Bread | Oil, honey, eggs, pumpkin purée, milk, pumpkin spice, baking soda, vanilla, salt, flour | 55-60 minutes |
| Turkey + Carrot Roll-Up | Tortilla, hummus, carrots, turkey lunch meat | 10 min |
| Strawberry Applause | Cherries, water, cinnamon, vanilla, strawberries | 15 minutes + mashing/pureeing time |

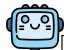 **Model Output of GPT4**:

| Recipe Name | Ingredients | Cooking Time |
|---|---|---|
| Pumpkin Bread | Oil, honey, eggs, pumpkin purée, milk, pumpkin spice, baking soda, vanilla, salt, flour, cinnamon | 55-60 minutes |
| Turkey + Carrot Roll-Up | Tortilla, hummus, carrots, turkey lunch meat | No cooking required |
| Strawberry Applause | Cherries, water, cinnamon, vanilla, strawberries | 30 minutes |

Figure 11: Case study on hard-level instruction with reasoning.

| Step | Prompt |
|---|---|
| Fixed Instruction Generation | I want to generate some real-world examples of information extraction that users would let the AI help with. Specifically, an example should contain the following two items: 1. Instruction: [User input, usually refers to extracting some desired information from a given text.] 2. Domain: [The domain to which the user query belongs.]

The following are several examples:
Example 1:
- Instruction: ...
- Domain: ...
Example 2:
- Instruction: ...
- Domain: ...

Following the format of the examples above, I would like you to help me generate ten more new examples that meet the following requirements: 1. These examples should be in various domains. 2. These examples should be described in different styles. 3. The generated domains do not overlap with the above example. |
| Background Text Generation | Give an information extraction instruction, we aim to generate some real-world example text from which the information can be extracted.
Specifically, each instruction includes the domain of background text and the type of extracted information.
So I would like the generated text to follow the domain targeted by the instruction and explicitly include the information that needs to be extracted.

The following are several examples:
Example 1:
- Instruction: ...
- Text: ...
Example 2:
- Instruction: ...
- Text: ...

Following the format of the examples above, I would like you to help me generate the text for the following instruction:
- Instruction: ... |
| Open Instruction Generation | Give a background text, generate an instruction which mentions extracting the information from it.
But don't point out what kind of information should be extracted.

The following are several examples:
Example 1:
- Text: ...
- Instruction: ...
Example 2:
- Text: ...
- Instruction: ...

Following the format of the examples above, I would like you to help me generate the instruction for the following text:
- Text: ... |
| Instruction Paraphrasing | Given ten instructions, paraphrase them one by one in different descriptive ways and make them like professional request but keep the key elements. So the outputs should ten paraphrased instructions. Remember not to output extra index or newline.
Sentence 1: ...
Sentence 2: ...
Sentence 3: ...
Sentence 4: ...
... |

Table 5: Prompts used for training data generation.

| Step | Prompt |
|------|--------|
| Table Generation (Direct) | Given an information extraction instruction and the background text, extract the information as a markdown table. If the instruction specifies the type of information to be extracted, ensure to follow the instruction. Otherwise, let this table include as many columns as possible. And keep the content brief.

The following are several examples:
Example 1:
- Instruction: ...
- Text: ...
- Table: ...
Example 2:
- Instruction: ...
- Text: ...
- Table: ...

Following the format of the examples above, I would like you to help me extract the table for the following instruction and text:
- Instruction: ...
- Text: ... |
| Table Generation (CoT) | Given an information extraction instruction and the background text, extract the information as a markdown table and produce a paragraph as the explanation. If the instruction specifies the type of information to be extracted, ensure to follow the instruction. Otherwise, let this table include as many columns as possible. And keep the content brief.

The following are several examples:
Example 1:
- Instruction: ...
- Text: ...
- Explanation: ...
- Table: ...
Example 2:
- Instruction: ...
- Text: ...
- Explanation: ...
- Table: ...

Following the format of the examples above, I would like you to help me extract the table for the below instruction and text. Please adopt a step-by-step approach: generate a comprehensive explanation as the first step, followed by table extraction as the second step:
- Instruction: ...
- Text: ... |

Table 6: Prompts for two methods of table generation: Direct and CoT.

| Models | Difficulty | | | Category | | Source | | Overall | # Data |
|--------|------|--------|------|-------|------|----------|----------|---------|--------|
| | Easy | Medium | Hard | Fixed | Open | Generate | Retrieve | | |
| **Open-Source Models** | | | | | | | | | |
| ALPACA | 27.42 | 27.61 | 25.19 | 30.90 | 15.29 | 26.80 | 27.16 | 27.11 | 52K |
| TÜLU | 29.45 | 29.80 | 30.90 | 33.20 | 18.62 | 27.98 | 30.65 | 30.07 | 512K |
| **Our Models** | | | | | | | | | |
| ODIE-DIRECT | **34.06** | **32.98** | **31.20** | **37.54** | 18.98 | **31.79** | **33.02** | **32.76** | 7.5K |
| ODIE-CoT | 28.70 | 26.09 | 27.99 | 29.81 | **21.13** | 25.24 | 28.33 | 27.60 | 7.1K |
| **Proprietary Models** | | | | | | | | | |
| CHATGPT | 35.03 | 28.98 | 32.71 | 35.04 | 23.29 | 29.22 | 33.11 | 32.21 | - |
| GPT-4 | 33.41 | 30.63 | 33.80 | 35.63 | 24.27 | 30.25 | 33.03 | 32.50 | - |

Table 7: Full results of table header evaluation. The metric is $F_1$ (%) of the exact matching score.

| Models | Difficulty | | | Category | | Source | | Overall | # Data |
|---|---|---|---|---|---|---|---|---|---|
| | Easy | Medium | Hard | Fixed | Open | Generate | Retrieve | | |
| **Open-Source Models** | | | | | | | | | |
| ALPACA | 64.66 | 59.31 | 54.53 | 65.89 | 45.69 | 59.57 | 59.90 | 59.80 | 52K |
| TÜLU | 74.18 | 66.73 | 67.02 | 77.78 | 49.26 | 69.39 | 69.47 | 69.44 | 512K |
| **Our Models** | | | | | | | | | |
| ODIE-DIRECT | 78.26 | **74.21** | **67.97** | **83.59** | 51.67 | 72.41 | **74.22** | **73.82** | 7.5K |
| - Filtering | **80.08** | 72.72 | 67.02 | 83.54 | 51.37 | **72.82** | 73.83 | 73.61 | 7.5K |
| ODIE-COT | 72.54 | 63.19 | 64.45 | 72.32 | **54.17** | 64.70 | 67.54 | 66.81 | 7.1K |
| - Filtering | 69.35 | 58.88 | 64.81 | 68.97 | 53.01 | 61.50 | 65.12 | 64.11 | 7.1K |
| **Proprietary Models** | | | | | | | | | |
| CHATGPT | 80.41 | 70.53 | 72.45 | 81.69 | 57.86 | 75.02 | 74.41 | 74.49 | - |
| GPT-4 | 77.77 | 70.58 | 75.51 | 82.06 | 57.78 | 78.36 | 73.29 | 74.47 | - |

Table 8: Full results of table header evaluation. The metric is $F_1$ (%) of the soft matching score.

| Models | Difficulty | | | Category | | Source | | Overall | # Data |
|---|---|---|---|---|---|---|---|---|---|
| | Easy | Medium | Hard | Fixed | Open | Generate | Retrieve | | |
| **Open-Source Models** | | | | | | | | | |
| ALPACA | 10.13 | 7.35 | 11.50 | 10.15 | 7.13 | 10.03 | 9.11 | 9.33 | 52K |
| TÜLU | 7.75 | 6.62 | 11.56 | 7.10 | 12.83 | 9.17 | 7.68 | 7.96 | 512K |
| **Our Models** | | | | | | | | | |
| ODIE-DIRECT | **17.74** | **15.35** | **14.28** | **15.59** | **16.19** | **13.83** | **16.42** | **15.80** | 7.5K |
| ODIE-COT | 14.22 | 10.98 | 10.73 | 11.41 | 13.39 | 11.64 | 11.96 | 11.90 | 7.1K |
| **Proprietary Models** | | | | | | | | | |
| CHATGPT | 16.11 | 16.38 | 12.96 | 15.34 | 14.76 | 12.47 | 16.13 | 15.23 | - |
| GPT-4 | 15.21 | 12.52 | 11.88 | 12.97 | 13.89 | 12.04 | 13.57 | 13.22 | - |

Table 9: Full results of table content evaluation. The metric is $F_1$ (%) of the exact matching score.

| Models | Difficulty | | | Category | | Source | | Overall | # Data |
|---|---|---|---|---|---|---|---|---|---|
| | Easy | Medium | Hard | Fixed | Open | Generate | Retrieve | | |
| **Open-Source Models** | | | | | | | | | |
| ALPACA | 52.09 | 40.00 | 49.57 | 48.32 | 41.73 | 48.08 | 46.45 | 46.86 | 52K |
| TÜLU | 66.77 | 60.05 | 63.50 | 64.57 | 57.83 | 65.41 | 62.34 | 63.14 | 512K |
| **Our Models** | | | | | | | | | |
| ODIE-DIRECT | **67.28** | **68.18** | **64.23** | **69.94** | 56.52 | 65.49 | **67.05** | **66.68** | 7.5K |
| ODIE-COT | 63.67 | 60.76 | 65.83 | 63.97 | 60.59 | 67.79 | 61.78 | 63.21 | 7.1K |
| **Proprietary Models** | | | | | | | | | |
| CHATGPT | 71.71 | 69.48 | 70.65 | 72.61 | 64.50 | 70.20 | 70.72 | 70.59 | - |
| GPT-4 | 75.70 | 73.09 | 75.99 | 76.67 | 69.13 | 76.08 | 74.45 | 74.83 | - |

Table 10: Full results of table content evaluation. The metric is $F_1$ (%) of the semantic similarity score.