# OpenReview forum: "Instruct and Extract: Instruction Tuning for On-Demand Information Extraction"
_EMNLP/2023/Conference — EMNLP 2023 Main_

### Official Review · Reviewer_i6Bg · 2023-08-03

**Soundness:** 3

**Excitement:**

3: Ambivalent: It has merits (e.g., it reports state-of-the-art results, the idea is nice), but there are key weaknesses (e.g., it describes incremental work), and it can significantly benefit from another round of revision. However, I won't object to accepting it if my co-reviewers champion it.

**Paper Topic And Main Contributions:**

This paper proposes a new task, On-Demand Information Extraction (ODIE), to fulfill the personalized needs of users. ODIE aims to extract the desired content and present it in a structured tabular format by following user instructions. The table headers are either user-specified (the "fixed" setting) or model-inferred (the "open" setting). A novel dataset named INSTRUCTIE is constructed, consisting of synthesized training data (14k) and human-annotated test data (150). The LLaMA-7B finetuned on the training data outperforms existing open-source models but is still inferior to proprietary models (ChatGPT and GPT-4). The main contributions lie in the novel task setting and the instruction-tuning dataset. However, it's not sure whether the dataset will be publicly available.


**Reasons To Accept:**

1. This paper focuses on an important research question: on-demand information extraction, which puts user demand ahead. The schemas of the extraction are explicitly specified or implicitly inferred from user instructions. This is a more realistic and applicable task setting in the era of large language models compared to the traditional IE settings where the schema is predefined.
2. This paper proposes a large instruction-tuning dataset to support and benchmark the ODIE task, including synthesized training data (14k) and human-annotated test data (150).


**Reasons To Reject:**

Though well-motivated, the task formulation is not well-defined and should be justified:
1. The output format is formulated as a relational table, where the headers are the target attributes and one row refers to one record. However, there are some output cases that this output format can't cover. For example, the multi-event extraction may outputs events of different event types, resulting in multiple relational tables. I suggest the authors clearly define the scope of the extraction or define a more flexible output format.
2. The cosine similarity (to evaluate the table header) and the ROUGE-L (to evaluate the table content) are both inappropriate. One intuition is that the relational tables are permutation-invariant for columns and rows, but neither of the two metrics is.
3. It's not sure whether the extracted items in the table content should be exact text pieces in the context (same as the traditional IE settings).

MINOR ASPECTS

1. The rating system of human evaluation is vague. For example, what are "correct"/"partly correct" table headers? If the "correct" means the exact match, there would be no need for manual assessment.
2. The 150 test samples are insufficient compared to the training set spanning a broad range of 83 domains.
3. Will the INSTRUCTIE dataset be publicly available?


**Reproducibility:**

3: Could reproduce the results with some difficulty. The settings of parameters are underspecified or subjectively determined; the training/evaluation data are not widely available.

**Reviewer Confidence:**

4: Quite sure. I tried to check the important points carefully. It's unlikely, though conceivable, that I missed something that should affect my ratings.

**Typos Grammar Style And Presentation Improvements:**

● Line 208: missing appendix.

● Line 250: what is a "style"?

● Line 295: Given -> given

● Line 441: thendescribe -> then describe

● Figure 4: missing the y-axis.

---

> ### Author Rebuttal · Authors · 2023-08-28
>
> Thank you for your thorough review and for recognizing the importance of putting user demand at the forefront in on-demand information extraction. We appreciate your positive remarks on the novel task setting and our efforts in constructing a dedicated instruction-tuning dataset to support and benchmark the ODIE task. We value your insights and have addressed the concerns you raised. Below, we provide a point-by-point response to the issues mentioned.
>
> **R1. About Task Format**
>
> We appreciate your feedback on the relational table format. We chose the relational table format as it is most intuitive to end-users and many standard IE tasks (such as NER and relation extraction) can be expressed in this format. This output format can also be used to represent ad-hoc user needs such as a combination of entities, entity attributes and relations, a task that would require a complex pipeline of IE components according to current task definitions.
>
> However we do acknowledge that our format does not encompass all current IE tasks in a convenient way. For event extraction with multiple event types, if the ontology is small, we can represent the output using a table with headers such as “event type, trigger, role 1, argument 1, role 2, argument 2…” and each row would be an event instance.  Another option would be to run our model multiple times, each with a different event type. We will mention this limitation in the paper and be more precise about the scope of our model.
>
>
>
> **R2. About Evaluation Metrics**
>
> Thanks for pointing out the concern.
>
> **Clarification on the adopted metrics**: We would like to clarify that the cosine similarity score in our paper is permutation-invariant for columns and rows. Because each table cell from the groundtruth first is aligned with one cell in the model outputs according to semantic similarity, and then we average the maximum similarity scores for all cells as the final scores. Meanwhile, ROUGE-L score is permutation-invariant for rows since it’s computed for each row. Similarly, each row from the groundtruth is aligned with one row in the model outputs for score calculation.
>
> **Supplementing more metrics**: Meanwhile, as recommended by other reviewers, we supplemented exact matching and semantic similarity (same as cosine similarity) as an evaluation metric for both header and content. The results validate our proposed model's commendable performance even with these stringent metrics.
>
> __(1) Exact matching for table header__
>
> | Models              | Difficulty |   |   | Category |  | Source Type |  | Overall    |
> |---------------------|------------------|--------------------|------------------|-------------------------|------------------------|------------------------|------------------------|------------|
> |               | **Easy** | **Medium** | **Hard** | **Fixed Header** | **Open Header** | **Generate** | **Retrieve** | **Overall**    |
> | Open-Source Models  |                  |                    |                  |                         |                        |                        |                        |             |
> | alpaca              | 27.42            | 27.61              | 25.19            | 30.90                   | 15.29                  | 26.80                  | 27.16                  | 27.11       |
> | TULU                | 29.45            | 29.80              | 30.90            | 33.20                   | 18.62                  | 27.98                  | 30.65                  | 30.07       |
> | **Our Models**          |                  |                    |                  |                         |                        |                        |                        |             |
> | **ODIE-direct**        | **34.06**            | **32.98**              | **31.20**            | **37.54**                   | **18.98**                  | **31.79**                  | **33.02**                  | **32.76**       |
> | **ODIE-cot**            | 28.70            | 26.09              | 27.99            | 29.81                   | 21.13                  | 25.24                  | 28.33                  | 27.60       |
> | Proprietary Models  |                  |                    |                  |                         |                        |                        |                        |             |
> | Turbo               | 35.03            | 28.98              | 32.71            | 35.04                   | 23.29                  | 29.22                  | 33.11                  | 32.21       |
> | GPT4                | 33.41            | 30.63              | 33.80            | 35.63                   | 24.27                  | 30.25                  | 33.03                  | 32.50       |
>
>
>
> __(2) Exact matching for table content__
> | Models              | Difficulty |   |   | Category |  | Source Type |  | Overall    |
> |---------------------|------------------|--------------------|------------------|-------------------------|------------------------|------------------------|------------------------|------------|
> |               | **Easy** | **Medium** | **Hard** | **Fixed Header** | **Open Header** | **Generate** | **Retrieve** | **Overall**    |
> | Open-Source Models  |                  |                    |                  |                         |                        |                        |                        |            |
> | Alpaca              | 10.13            | 7.35               | 11.50            | 10.15                   | 7.13                   | 10.03                  | 9.11                   | 9.33       |
> | Tulu                | 7.75             | 6.62               | 11.56            | 7.10                    | 12.83                  | 9.17                   | 7.68                   | 7.96       |
> | **Our Models**          |                  |                    |                  |                         |                        |                        |                        |            |
> | **ODIE-Direct**         | **17.74**            | **15.35**              | **14.28**            | **15.59**                   | **16.19**                  | **13.83**                  | **16.42**                  | **15.80**      |
> | **ODIE-CoT**            | 14.22            | 10.98              | 10.73            | 11.41                   | 13.39                  | 11.64                  | 11.96                  | 11.90      |
> | Proprietary Models  |                  |                    |                  |                         |                        |                        |                        |            |
> | turbo               | 16.11            | 16.38              | 12.96            | 15.34                   | 14.76                  | 12.47                  | 16.13                  | 15.23      |
> | GPT4                | 15.21            | 12.52              | 11.88            | 12.97                   | 13.89                  | 12.04                  | 13.57                  | 13.22      |
>
>
> __(3) Semantic similarity for table content__
> | Models              | Difficulty |   |   | Category |  | Source Type |  | Overall    |
> |---------------------|------------------|--------------------|------------------|-------------------------|------------------------|------------------------|------------------------|------------|
> |               | **Easy** | **Medium** | **Hard** | **Fixed Header** | **Open Header** | **Generate** | **Retrieve** | **Overall**    |
> | Open-Source Models  |                  |                    |                  |                         |                        |                        |                        |            |
> | Alpaca              | 52.09            | 40.00              | 49.57            | 48.32                   | 41.73                  | 48.08                  | 46.45                  | 46.86      |
> | Tulu                | 66.77            | 60.05              | 63.50            | 64.57                   | 57.83                  | 65.41                  | 62.34                  | 63.14      |
> | **Our Models**          |                  |                    |                  |                         |                        |                        |                        |            |
> | **ODIE-Direct**         | **67.28**            | **68.18**             | 64.23            | **69.94**                   | 56.52                  | 65.49                  | **67.05**                  | **66.68**      |
> | **ODIE-CoT**            | 63.67            | 60.76              | **65.83**            | 63.97                   | **60.59**                  | **67.79**                  | 61.78                  | 63.21      |
> | Proprietary Models  |                  |                    |                  |                         |                        |                        |                        |            |
> | turbo               | 71.71            | 69.48              | 70.65            | 72.61                   | 64.50                  | 70.20                  | 70.72                  | 70.59      |
> | GPT4                | 75.70            | 73.09              | 75.99            | 76.67                   | 69.13                  | 76.08                  | 74.45                  | 74.83      |
>
>
> __Correlation Analysis__: To further study different metrics on our proposed task, we analyze the correlation between these automatic metrics (Exact Match, Semantics Similarity, ROUGEL) and human evaluations. The analysis results are listed below, including the Pearson, Spearman, and Kendall coefficients. The results indicate that, for table header, both of these three metrics are highly correlated with human evaluation. But for table content,  semantic similarity and RougeL can indeed be more reliable metrics for this task compared with exact matching.
>
> | Metrics     | Pearson  | Spearman | Kendall  |
> |------------------|----------|----------|----------|
> | [Header] Exact Matching      | 0.640    | 0.609    | 0.494    |
> | [Header] Semantic Similarity      | **0.817**    | **0.769**    | **0.637**    |
> | [Header] RougeL    | 0.694    | 0.735    | 0.608    |
> | [Content] Exact Matching     | 0.338    | 0.375    | 0.272    |
> | [Content] Semantic Similarity     | **0.764**    | **0.705**    | **0.558**    |
> | [Content] RougeL   | 0.713    | 0.704    | 0.554    |
>
> We sincerely appreciate the reviewers for highlighting the evaluation issue, allowing us to conduct a thorough study that provides more detailed insights and resolves ambiguities stemming from the use of a single metric. These new findings will be definitely added in the next paper version. We will be happy to discuss at openreview platform if the reviewers have further questions or comments.
> As mentioned in the limitation section, we admit the challenges of proposing comprehensive evaluation metrics for on-demand information extraction. Proposing better metrics is a critical aspect that we aim to explore in our future work.
>
>
>
>
> **R3. Table contents being exact text pieces**
>
> In our problem setting, we did not enforce that table contents must be spans of the original text. Our instructions (especially those in the hard set) require some amount of reasoning and summarization over the text so the final result might not be available as a text span.
>
>
>
> **R4. Rating System in Human Evaluation**
>
> We understand the need for clarity in this aspect. In our setting, "correct" and "partly correct" are not strictly bound to exact matches. We allow the extracted contents to be semantically aligned with the original text, acknowledging that different phrasings might convey equivalent meanings.
> For example, it would be regarded as correct if the groundtruth is “2 min” but the model output is “2 minutes”.  Because we think capturing the semantic essence is flexible and paramount compared with rigid exact matches, especially in diverse real-world scenarios.
>
>
>
>
> **R5. Size of the Test Set**
>
> Thanks for raising your concern. Our 150 test instances involve 61 various domains. These instances are both representative and resonate with the genuine requirements of varied user groups. Additionally (as mentioned in LINES 403-414), the scale of our test set aligns with existing instruction-tuning benchmarks, such as 129 in Anthropic, 156 in the Koala evaluation, and 188 in Open Assistant.
>
>
> **R6. Availability of Dataset**
>
> We have submitted the dataset as Supplementary Materials, along with different model outputs and human evaluation results.  Please feel free to take a look if you are interested. Also, the data, model outputs, and human evaluations will be open-sourced after notification.

---

### Official Review · Reviewer_Z3zM · 2023-08-04

**Soundness:** 4

**Excitement:**

4: Strong: This paper deepens the understanding of some phenomenon or lowers the barriers to an existing research direction.

**Paper Topic And Main Contributions:**

The paper seeks to democratize information extraction (IE), addressing
a task it calls "on-demand information extraction," essentially a
zero-shot document-level table construction task based on textual
prompts provided by putative end users.  The paper describes a system
capable of performing two variants of this task, one in which the
extraction targets (fields or columns) are specified, and one in which
they are not.  In development of this system, the paper creates a new
data source of synthetic extraction tasks spanning a large number of
domains and describes the process by which it is used for instruction
training of a small model in the GPT family (Llama 7B).  Extensive
experiments benchmark the system against a range of alternatives in
the same family, showing that it outperforms models not trained in the
same fashion, including much larger models--although performance tends
to lack behind ChatGPT and GPT4.


**Questions For The Authors:**

A. You make much of the multi-domain aspect of the experimental data.
How are domains defined?  I believe you explicitly specify the domain in
generating synthetic data.  Is this the only domain "knob" you employ?

B. Could you say more about the retrieval of real texts and how those
were sorted by domain?

Cl Your description of the human evaluation is a little vague.  I'm
inferring that a complete datapoint in that evaluation was a complete
model output (all fields and extractions).  Does "correct and
satisfying" mean the model got all headers and fields exactly correct,
or just "correct enough for my purposes"?


**Reasons To Accept:**

I think the contributions of the paper are substantial.  Early
evidence seems to suggest that instruction-trained LLMs are not
particularly strong in IE tasks, one possible reason being that they
haven't been exposed to tasks of the right type.  The paper tends to
confirm this hypothesis and offers a replicable procedure (including a
new dataset of IE instructions?) to correct this shortcoming.

The paper is well written, with good separation between paper body and
appendix.  The training procedure nicely addresses difficulties that
can be encountered in working with synthetic text as training data, a
procedure that is clearly described.  It should be possible for other
researchers to replicate the effects observed in the paper (if not the
specific outcomes) and build on this research.  I believe that many
will want to do so.

It is appealing that the work was done with a "small"
instruction-trained LLM.  An ongoing pain point in NLP research is the
reliance on proprietary models for some of the more interesting
research directions.  Llama 7B can be run on a beefy laptop, making it
feasible for the community to adapt and extend this work.


**Reasons To Reject:**

Although I appreciate the paper's motivation to provide the benefits
of IE to end users, I don't really see the need to advocate for a new
task.  I have difficulty imagining scenarios in which end users would
want to use such a capability.  IE is typically a means to many
different ends, making technicians the main constituency likely to be
interested in improvements to IE technology.  Thus, I think the
central emphasis of the paper is off-target.  This is notable work, in
my opinion mainly because it helps the field out of ruts it's become
too comfortable with, with a heavy focus on particular IE domains over
others and a conceptualization (NER, relation extraction, event
extraction) that made sense initially, but may now be an impediment to
progress.  The paper offers a direction that enables rapid deployment
of IE to underserved domains, of which there is probably an infinite
supply.

Because of this perspective, I found the performance evaluations
somewhat underwhelming, with their use of soft matching, SBERT
similarity, ROUGE, etc.  These metrics make sense for more qualitative
tasks, where goodness is difficult to characterize precisely, but not
so much for IE.  (It's probably reasonable to evaluate the open
variant of the paper's task, which I do find interesting, in such a
fashion.) The paper acknowledges this shortcoming in the Limitations
section (which is thoughtfully written).  I think this flaw is
material.  It is really hard to tell how this work moves the needle
with respect to the baseline of supervised extraction.  I think it
does, but have no good evidence for that opinion.

The paper doesn't offer a lot of ideas about where to go next.  What
are useful directions, now that we have a model like this?  Can you
improve its accuracy somehow?  Or are you heading in the direction of
applications, as implied by the user-facing task you define.  There
are a couple "future work" ideas masquerading as Limitations, such as
more rigorous accuracy evaluations and some kind of interleaving of
direct and CoT approaches.  These ideas and any other next steps merit
mention in the body of the paper.


**Reproducibility:**

3: Could reproduce the results with some difficulty. The settings of parameters are underspecified or subjectively determined; the training/evaluation data are not widely available.

**Reviewer Confidence:**

4: Quite sure. I tried to check the important points carefully. It's unlikely, though conceivable, that I missed something that should affect my ratings.

**Typos Grammar Style And Presentation Improvements:**

A more succinct statement of contributions would be helpful.  If you
intend to release InstructIE, so state.  I suspect it would excite
interest.

Should the (b) label in Figure 5 read "Evaluation for table contents"?

---

> ### Author Rebuttal · Authors · 2023-08-28
>
> Thank you for your thoughtful review, which recognizes the importance of our proposed task. We appreciate your acknowledgment of the replicability of our training procedures and the potential for broader community engagement. Your comments on the accessibility of using a smaller model like Llama 7B are particularly encouraging, and we're pleased you see the value in that. We value your insights and have addressed the concerns you raised. Below, we provide a point-by-point response to the issues mentioned.
>
> **R1. About application scenarios**
>
> We deeply appreciate your acknowledgment of our motivation. We think the proposed task, On-Demand Information Extraction, has the vast applicability across industries and its potential to streamline tasks, making information extraction more efficient and user-centric.
>
> For example,  doctors can pinpoint needed health metrics from a patient's history, like "medications and dosages over the last year for the patient," for quicker, more targeted information. Also, legal professionals can speed up reviews by specifying queries like "extract clauses on breach of the contract penalties."
>
> In these real world scenarios, existing IE approaches cannot be directly applied to and handle them because these approaches tend to extract exhaustively and rely on predefined ontologies. But equipped with our setting, an average user can just input a natural language instruction to define and extract the desired information from texts.
>
>
>
>
> **R2. About Evaluation Metrics**
>
> We sincerely appreciate the reviewer for highlighting the evaluation issue, allowing us to conduct a thorough study that provides more detailed insights and resolves ambiguities stemming from the use of a single metric.
>
> Specifically, we supplemented exact matching and semantic similarity (same as cosine similarity) as an evaluation metric for both header and content, as recommended by other reviewers. The results validate our proposed model's commendable performance even with these stringent metrics.
>
> __(1) Exact matching for table header__
>
> | Models              | Difficulty |   |   | Category |  | Source Type |  | Overall    |
> |---------------------|------------------|--------------------|------------------|-------------------------|------------------------|------------------------|------------------------|------------|
> |               | **Easy** | **Medium** | **Hard** | **Fixed Header** | **Open Header** | **Generate** | **Retrieve** | **Overall**    |
> | Open-Source Models  |                  |                    |                  |                         |                        |                        |                        |             |
> | alpaca              | 27.42            | 27.61              | 25.19            | 30.90                   | 15.29                  | 26.80                  | 27.16                  | 27.11       |
> | TULU                | 29.45            | 29.80              | 30.90            | 33.20                   | 18.62                  | 27.98                  | 30.65                  | 30.07       |
> | **Our Models**          |                  |                    |                  |                         |                        |                        |                        |             |
> | **ODIE-direct**        | **34.06**            | **32.98**              | **31.20**            | **37.54**                   | **18.98**                  | **31.79**                  | **33.02**                  | **32.76**       |
> | **ODIE-cot**            | 28.70            | 26.09              | 27.99            | 29.81                   | 21.13                  | 25.24                  | 28.33                  | 27.60       |
> | Proprietary Models  |                  |                    |                  |                         |                        |                        |                        |             |
> | Turbo               | 35.03            | 28.98              | 32.71            | 35.04                   | 23.29                  | 29.22                  | 33.11                  | 32.21       |
> | GPT4                | 33.41            | 30.63              | 33.80            | 35.63                   | 24.27                  | 30.25                  | 33.03                  | 32.50       |
>
>
>
> __(2) Exact matching for table content__
> | Models              | Difficulty |   |   | Category |  | Source Type |  | Overall    |
> |---------------------|------------------|--------------------|------------------|-------------------------|------------------------|------------------------|------------------------|------------|
> |               | **Easy** | **Medium** | **Hard** | **Fixed Header** | **Open Header** | **Generate** | **Retrieve** | **Overall**    |
> | Open-Source Models  |                  |                    |                  |                         |                        |                        |                        |            |
> | Alpaca              | 10.13            | 7.35               | 11.50            | 10.15                   | 7.13                   | 10.03                  | 9.11                   | 9.33       |
> | Tulu                | 7.75             | 6.62               | 11.56            | 7.10                    | 12.83                  | 9.17                   | 7.68                   | 7.96       |
> | **Our Models**          |                  |                    |                  |                         |                        |                        |                        |            |
> | **ODIE-Direct**         | **17.74**            | **15.35**              | **14.28**            | **15.59**                   | **16.19**                  | **13.83**                  | **16.42**                  | **15.80**      |
> | **ODIE-CoT**            | 14.22            | 10.98              | 10.73            | 11.41                   | 13.39                  | 11.64                  | 11.96                  | 11.90      |
> | Proprietary Models  |                  |                    |                  |                         |                        |                        |                        |            |
> | turbo               | 16.11            | 16.38              | 12.96            | 15.34                   | 14.76                  | 12.47                  | 16.13                  | 15.23      |
> | GPT4                | 15.21            | 12.52              | 11.88            | 12.97                   | 13.89                  | 12.04                  | 13.57                  | 13.22      |
>
>
> __(3) Semantic similarity for table content__
> | Models              | Difficulty |   |   | Category |  | Source Type |  | Overall    |
> |---------------------|------------------|--------------------|------------------|-------------------------|------------------------|------------------------|------------------------|------------|
> |               | **Easy** | **Medium** | **Hard** | **Fixed Header** | **Open Header** | **Generate** | **Retrieve** | **Overall**    |
> | Open-Source Models  |                  |                    |                  |                         |                        |                        |                        |            |
> | Alpaca              | 52.09            | 40.00              | 49.57            | 48.32                   | 41.73                  | 48.08                  | 46.45                  | 46.86      |
> | Tulu                | 66.77            | 60.05              | 63.50            | 64.57                   | 57.83                  | 65.41                  | 62.34                  | 63.14      |
> | **Our Models**          |                  |                    |                  |                         |                        |                        |                        |            |
> | **ODIE-Direct**         | **67.28**            | **68.18**             | 64.23            | **69.94**                   | 56.52                  | 65.49                  | **67.05**                  | **66.68**      |
> | **ODIE-CoT**            | 63.67            | 60.76              | **65.83**            | 63.97                   | **60.59**                  | **67.79**                  | 61.78                  | 63.21      |
> | Proprietary Models  |                  |                    |                  |                         |                        |                        |                        |            |
> | turbo               | 71.71            | 69.48              | 70.65            | 72.61                   | 64.50                  | 70.20                  | 70.72                  | 70.59      |
> | GPT4                | 75.70            | 73.09              | 75.99            | 76.67                   | 69.13                  | 76.08                  | 74.45                  | 74.83      |
>
>
> __Correlation Analysis__: To further study different metrics on our proposed task, we analyze the correlation between these automatic metrics (Exact Match, Semantic Similarity, ROUGEL) and human evaluations. The analysis results are listed below, including the Pearson, Spearman, and Kendall coefficients. The results indicate that, for table header, these three metrics are highly correlated with human evaluation. But for table content, semantic similarity and RougeL can indeed be more reliable metrics for this task compared with exact matching.
>
> | Metrics     | Pearson  | Spearman | Kendall  |
> |------------------|----------|----------|----------|
> | [Header] Exact Matching      | 0.640    | 0.609    | 0.494    |
> | [Header] Semantic Similarity      | **0.817**    | **0.769**    | **0.637**    |
> | [Header] RougeL    | 0.694    | 0.735    | 0.608    |
> | [Content] Exact Matching     | 0.338    | 0.375    | 0.272    |
> | [Content] Semantic Similarity     | **0.764**    | **0.705**    | **0.558**    |
> | [Content] RougeL   | 0.713    | 0.704    | 0.554    |
>
>
> These new findings will be definitely added in the next paper version. We will be happy to discuss at openreview platform if the reviewers have further questions or comments.
>
> As mentioned in the limitation section, we admit the challenges of proposing comprehensive evaluation metrics for on-demand information extraction. Proposing better metrics is a critical aspect that we aim to explore in our future work.
>
>
>
>
> **R3. About Future Direction**
>
> Thanks for pointing out these questions. We think there are at least 5 promising directions based on this work (three of them mentioned in the limitation section).
> *Hybrid Method Combination*: We've assessed Direct and CoT separately but haven't studied their synergistic effects, which could enhance performance and provide new insights.
> *textbf{Evaluation Metrics*: Current metrics focus on similarity to ground truth. A more granular set of metrics is needed for table quality and organization.
> *Contextual Inference and Complex Instructions*: Room exists for improvement in contextual understanding and handling intricate queries, to make the system more accessible.
> *Exploration of Complex Output Formats*: Traditional methods use fixed output structures; investigating more complex and flexible formats can cater to varied user needs.
> *Integration with Other Systems: On-Demand IE can enhance other systems, such as database management tools, and virtual assistants. Future work should target seamless integration via standardized APIs and performance tweaks.
>
>
>
> **R4. About Choosing Domain**
>
> We're glad you appreciate the diversity of our dataset. For the training data, domains are autonomously generated by ChatGPT in separate batches, with an emphasis on ensuring each batch contains distinct domains for wide-ranging coverage (as mentioned in LINES 209-221). As for the test data, leveraging GPT-4's vast knowledge, we initially generated a pool of 1,000 instructions spanning diverse domains. From this, annotators meticulously selected 150 samples that are not only representative but also align with the genuine needs of different user groups. These domain labels are further revised and merged considering label semantics. (as described in LINES 319-330).
>
>
>
> **R5. About Retrieval of Real Texts**
>
> Given a domain, we first manually identified a representative type of text for that domain, like legal documents for law. Using a search engine like Newbing, we accessed the desired text type, ensuring they contained ample and suitable information elements for extraction. We also ensured text lengths between 100 and 1,000 words to meet the large language models' input length restrictions.
>
>
> **R6. About Human Evaluation**
>
> In our human evaluation, "correct and satisfying" implies that the extracted information was both semantically accurate and aligned with the user's query. It doesn't necessitate an exact match for every header and field but ensures the output effectively meets the user's intent. For example, it would be regarded as correct if the groundtruth is “2 min” but the model output is “2 minutes”.  Because we think capturing the semantic essence is flexible and paramount compared with rigid exact matches, especially in diverse real-world scenarios.
>
> **Reference**
>
> [1] Yizhong Wang*, Swaroop Mishra*, Pegah Alipoormolabashi, Yeganeh Kordi et al.Super-NaturalInstructions: Generalization via Declarative Instructions on 1600+ NLP Tasks. EMNLP 2022.
>
> [2] Yizhong Wang, Yeganeh Kordi, Swaroop Mishra, Alisa Liu, Noah A. Smith, Daniel Khashabi, and Hannaneh Hajishirzi. 2022a. Self-instruct: Aligning language model with self generated instructions. ACL 2023.
>
> [3] Nils Reimers and Iryna Gurevych. 2019. Sentence-bert: Sentence embeddings using siamese bert-networks. EMNLP-IJCNLP 2019.

---

### Official Review · Reviewer_REDP · 2023-08-11

**Soundness:** 3

**Excitement:**

3: Ambivalent: It has merits (e.g., it reports state-of-the-art results, the idea is nice), but there are key weaknesses (e.g., it describes incremental work), and it can significantly benefit from another round of revision. However, I won't object to accepting it if my co-reviewers champion it.

**Paper Topic And Main Contributions:**

This paper proposes to curate instruction tuning resources and train a instruction following model to perform “on-demand information extraction“ task (which is a table extraction task with a dynamic table head requirement). The authors propose an automatic pipeline to generate (instruction, text, table) data points for instruction tuning, and tune a LLaMA model using the newly generated InstructIE dataset. They also propose a manually curated evaluation dataset with 150 samples to evaluate the performance in terms of head and table content. Evaluation results show that the proposed model achieves improvement compared with open-source LLM, which is the base model before fine-tuning on the new instruction tuning dataset.

**Questions For The Authors:**

- As mentioned in line 232-243, the open instructions are generated given the background text, would this setting guarantee most of the fields has related information mentioned in the background text. Are they any arrangement to make sure the instruction tuning dataset contains text with N/A ground-truth structure?
- What would be the performance of supervised models that are directly fine-tuned on table extraction datasets? It would be nice to have a comparison to provide a better understanding of the capability of zero-shot LLM.
- It’s great to see the proposed dataset covers a lot of domains. How are those domains annotated? Is the domain categorization from some existing ontology/definition?

**Reasons To Accept:**

- Task formulation, resources and modeling for a more flexible information extraction formulation are quite needed
- Automatically generated instruction tuning dataset and manually created test set with 150 instances would be helpful
- Empirical results show that tuning on the generated dataset would improve the IE results on the table extraction evaluation dataset

**Reasons To Reject:**

- The task formulation of “dynamic table extraction with the more flexible header“is not quite matched with the claim of “on-demand information extraction“, it would be necessary to define the scope better
- Not sure how much gain is from instruction tuning compared to simply fine-tuning with task-specific data points without instruction info. As we can expect fine-tuning a LM on task data (even without instruction) can still lead to some performance improvement. A comparison with fine-tuning on task instances without instruction would be helpful to quantify such a gain
- Using ROUGE score to evaluate IE results leads to an unsatisfactory setting. As the legal values of certain table fields could be very similar, but incorrect. If the exact match results could be reported it would be very helpful to clear such ambiguity.

**Reproducibility:**

4: Could mostly reproduce the results, but there may be some variation because of sample variance or minor variations in their interpretation of the protocol or method.

**Reviewer Confidence:**

5: Positive that my evaluation is correct. I read the paper very carefully and I am very familiar with related work.

---

> ### Author Rebuttal · Authors · 2023-08-28
>
> Thank you for your positive review and for acknowledging the importance of our task formulation, resource generation, and empirical results. We're encouraged that you find our approach and datasets to be valuable contributions to the field of information extraction. Your feedback affirms the utility and effectiveness of our model and methodology. We appreciate the opportunity to clarify and address the concerns you've raised. Below, we provide a point-by-point response to the issues mentioned.
>
> **R1. About Task Formulation**
>
> Thank you for raising your concern. By claiming on-demand information extraction, we aim to equip the average users with the ability to extract the desired information by specifying a natural language instruction.
> Regarding the intention behind a table output format, it’s indeed to present extracted information in a manner that's immediately accessible and comprehensible to end-users. The users can customize the table based on the type of information they seek to fulfill their demands. This eliminates the constraints of traditional systems that abide by a rigid format.
> However, we still admit the current task formulation still has room for improvement in processing more complex structures as the extracted output for fulfilling more challenging user needs.
>
> **R2. About Instruction Tuning vs. Fine-Tuning**
>
> We appreciate this perspective and conducted an experiment accordingly. Specifically, we train the LLaMA 7b model without any specific user instructions and provide just a general system instruction, “extract a table from the text”. To ensure the comprehensive evaluation, we evaluate the model outputs on five different metrics, including two for table header and three for table content (please find more explanations about evaluation metrics in the third answer). The experimental results are listed below in the table.  It shows that the direct version of our full model significantly outperforms the variant trained without user instructions, which indicates distinct advantages of instruction tuning. It provides the model with explicit prompts about the desired output structure, enabling better alignment with user requirements like customized table headers or specific expected table content.
>
> | Method          | Header  |   | Content |     |     |
> |-----------------|------------------------|------------------------|-----------------|-------------------------|-------------------------|
> |     |  **Exact matching**  |  **Semantic Similarity** | **Exact matching** | **Semantic Similarity** | **RougeL** |
> | **ODIE**            | **32.76**                   | **73.82**                   | **15.80**           | **66.68**                    | **45.93**                    |
> | w/o instruction | 28.17                   | 62.35                   | 7.53            | 53.72                    | 26.90                    |
>
>
> **R3. About Evaluation Metrics**
>
> __Initial Choice of ROUGE-L__: ROUGE-L was initially selected for its ability to measure n-gram overlap, offering a more flexible evaluation than exact-match metrics. Furthermore, its wide use in previous instruction-tuning works [1, 2] also ensures our work is comparable to prior research.
>
> __Incorporating two new metrics__: To address this limitation, we adopt two more metrics, which are exact matching and semantic similarity , alongside ROUGE scores in our evaluation. Specifically, exact matching requires two table cells to be exactly the same while semantics similarity uses SentenceBERT [3] to calculate the cosine similarity as the semantic similarity score between two cells. Both metrics are micro F1 scores.
>
> __Experimental results__: The results are shown below in the three tables for both table header and content (Semantics similarity for table headers has been included in our submission). The results demonstrate that our proposed model continues to achieve superior performance under these new metrics.
>
> __(1) Exact matching for table header__
>
> | Models              | Difficulty |   |   | Category |  | Source Type |  | Overall    |
> |---------------------|------------------|--------------------|------------------|-------------------------|------------------------|------------------------|------------------------|------------|
> |               | **Easy** | **Medium** | **Hard** | **Fixed Header** | **Open Header** | **Generate** | **Retrieve** | **Overall**    |
> | Open-Source Models  |                  |                    |                  |                         |                        |                        |                        |             |
> | alpaca              | 27.42            | 27.61              | 25.19            | 30.90                   | 15.29                  | 26.80                  | 27.16                  | 27.11       |
> | TULU                | 29.45            | 29.80              | 30.90            | 33.20                   | 18.62                  | 27.98                  | 30.65                  | 30.07       |
> | **Our Models**          |                  |                    |                  |                         |                        |                        |                        |             |
> | **ODIE-direct**        | **34.06**            | **32.98**              | **31.20**            | **37.54**                   | **18.98**                  | **31.79**                  | **33.02**                  | **32.76**       |
> | **ODIE-cot**            | 28.70            | 26.09              | 27.99            | 29.81                   | 21.13                  | 25.24                  | 28.33                  | 27.60       |
> | Proprietary Models  |                  |                    |                  |                         |                        |                        |                        |             |
> | Turbo               | 35.03            | 28.98              | 32.71            | 35.04                   | 23.29                  | 29.22                  | 33.11                  | 32.21       |
> | GPT4                | 33.41            | 30.63              | 33.80            | 35.63                   | 24.27                  | 30.25                  | 33.03                  | 32.50       |
>
>
>
> __(2) Exact matching for table content__
> | Models              | Difficulty |   |   | Category |  | Source Type |  | Overall    |
> |---------------------|------------------|--------------------|------------------|-------------------------|------------------------|------------------------|------------------------|------------|
> |               | **Easy** | **Medium** | **Hard** | **Fixed Header** | **Open Header** | **Generate** | **Retrieve** | **Overall**    |
> | Open-Source Models  |                  |                    |                  |                         |                        |                        |                        |            |
> | Alpaca              | 10.13            | 7.35               | 11.50            | 10.15                   | 7.13                   | 10.03                  | 9.11                   | 9.33       |
> | Tulu                | 7.75             | 6.62               | 11.56            | 7.10                    | 12.83                  | 9.17                   | 7.68                   | 7.96       |
> | **Our Models**          |                  |                    |                  |                         |                        |                        |                        |            |
> | **ODIE-Direct**         | **17.74**            | **15.35**              | **14.28**            | **15.59**                   | **16.19**                  | **13.83**                  | **16.42**                  | **15.80**      |
> | **ODIE-CoT**            | 14.22            | 10.98              | 10.73            | 11.41                   | 13.39                  | 11.64                  | 11.96                  | 11.90      |
> | Proprietary Models  |                  |                    |                  |                         |                        |                        |                        |            |
> | turbo               | 16.11            | 16.38              | 12.96            | 15.34                   | 14.76                  | 12.47                  | 16.13                  | 15.23      |
> | GPT4                | 15.21            | 12.52              | 11.88            | 12.97                   | 13.89                  | 12.04                  | 13.57                  | 13.22      |
>
>
> __(3) Semantic similarity for table content__
> | Models              | Difficulty |   |   | Category |  | Source Type |  | Overall    |
> |---------------------|------------------|--------------------|------------------|-------------------------|------------------------|------------------------|------------------------|------------|
> |               | **Easy** | **Medium** | **Hard** | **Fixed Header** | **Open Header** | **Generate** | **Retrieve** | **Overall**    |
> | Open-Source Models  |                  |                    |                  |                         |                        |                        |                        |            |
> | Alpaca              | 52.09            | 40.00              | 49.57            | 48.32                   | 41.73                  | 48.08                  | 46.45                  | 46.86      |
> | Tulu                | 66.77            | 60.05              | 63.50            | 64.57                   | 57.83                  | 65.41                  | 62.34                  | 63.14      |
> | **Our Models**          |                  |                    |                  |                         |                        |                        |                        |            |
> | **ODIE-Direct**         | **67.28**            | **68.18**             | 64.23            | **69.94**                   | 56.52                  | 65.49                  | **67.05**                  | **66.68**      |
> | **ODIE-CoT**            | 63.67            | 60.76              | **65.83**            | 63.97                   | **60.59**                  | **67.79**                  | 61.78                  | 63.21      |
> | Proprietary Models  |                  |                    |                  |                         |                        |                        |                        |            |
> | turbo               | 71.71            | 69.48              | 70.65            | 72.61                   | 64.50                  | 70.20                  | 70.72                  | 70.59      |
> | GPT4                | 75.70            | 73.09              | 75.99            | 76.67                   | 69.13                  | 76.08                  | 74.45                  | 74.83      |
>
>
> __Correlation Analysis__: To further study different metrics on our proposed task, we analyze the correlation between these automatic metrics (Exact Match, Semantic Similarity, ROUGEL) and human evaluations. The analysis results are listed below, including the Pearson, Spearman, and Kendall coefficients. The results indicate that, for table header, these three metrics are highly correlated with human evaluation. But for table content, semantic similarity and RougeL can indeed be more reliable metrics for this task compared with exact matching.
>
> | Metrics     | Pearson  | Spearman | Kendall  |
> |------------------|----------|----------|----------|
> | [Header] Exact Matching      | 0.640    | 0.609    | 0.494    |
> | [Header] Semantic Similarity      | **0.817**    | **0.769**    | **0.637**    |
> | [Header] RougeL    | 0.694    | 0.735    | 0.608    |
> | [Content] Exact Matching     | 0.338    | 0.375    | 0.272    |
> | [Content] Semantic Similarity     | **0.764**    | **0.705**    | **0.558**    |
> | [Content] RougeL   | 0.713    | 0.704    | 0.554    |
>
>
> We sincerely appreciate the reviewers for highlighting the evaluation issue, allowing us to conduct a thorough study that provides more detailed insights and resolves ambiguities stemming from the use of a single metric. These new findings will be definitely added in the next paper version. We will be happy to discuss at the platform if the reviewers have further questions or comments.
> As mentioned in the limitation section, we admit the challenges of proposing comprehensive evaluation metrics for on-demand information extraction. Proposing better metrics is a critical aspect that we aim to explore in our future work.
>
>
>
> **R4. About Empty Cell for Open Instructions**
>
> Thank you for this query.
>
> An open instruction does not specify the type of header, i.e., “position” and “salary”, but rather employs relatively vague requirements, such as “key information” (as mentioned in Lines 235-239). For such an instruction, the model aims for an output inclusive of all relevant details from the input text  (Lines 260-264). Thus, most fields derive directly from the text. That said, we do allow tables to have empty cells for instances where certain information might be missing from the input (Lines 286-289).  According to statistics of our test data, there are a total of 1950 table cells, of which 103 are N/A, making up 5% of the total.
>
> To guarantee that the groundtruth tables are informative, we have implemented filtering mechanisms for both the generated training and annotated test data. This ensures that tables with excessive empty cells (indicative of missed information) are not part of our dataset, as detailed in Lines 286-289.
>
>
>
>
> **R5. Compare with supervised table extraction method**
>
> Thanks for asking this question. We further conduct experiments on a supervised table extraction approach, text2table [4], from ACL 2022. Specifically, we adopt the public implementation of this bart-based model trained with the default setting on the WikiTableText dataset and tested with table constraints on our test data. The experiment results are shown below, which demonstrate our method obviously outperforms this baseline. We think it’s because our method can benefit from customized table structures and contents following flexible user instructions.
>
> | Method          | Header  |   | Content |     |
> |-----------------|------------------------|------------------------|-----------------|-------------------------|
> |     |  **Exact matching**  |  **Semantic Similarity** | **Exact matching** | **Semantic Similarity** |
> | **ODIE**            | **32.76**                   | **73.82**                   | **15.80**           | **66.68**                    |
> | Text2table | 15.02                   | 39.73                   | 2.01            | 13.32                    |
>
>
>
> **R6. About Domain Annotations**
>
> We're glad you appreciate the diversity of our dataset. For the training data, domains are autonomously generated by ChatGPT in separate batches, with an emphasis on ensuring each batch contains distinct domains for wide-ranging coverage (Lines 209-221). As for the test data, leveraging GPT-4's vast knowledge, we initially generated a pool of 1,000 instructions spanning diverse domains. From this, annotators meticulously selected 150 samples that are not only representative but also align with the genuine needs of different user groups (as described in Lines 319-330).
>
>
> **Reference**:
>
> [1] Yizhong Wang*, Swaroop Mishra*, Pegah Alipoormolabashi, Yeganeh Kordi et al.Super-NaturalInstructions: Generalization via Declarative Instructions on 1600+ NLP Tasks. EMNLP 2022.
>
> [2] Yizhong Wang, Yeganeh Kordi, Swaroop Mishra, Alisa Liu, Noah A. Smith, Daniel Khashabi, and Hannaneh Hajishirzi. 2022a. Self-instruct: Aligning language model with self generated instructions. ACL 2023.
>
> [3] Nils Reimers and Iryna Gurevych. 2019. Sentence-bert: Sentence embeddings using siamese bert-networks. EMNLP-IJCNLP 2019.
>
> [4] Xueqing Wu, Jiacheng Zhang, Hang Li. Text-to-Table: A New Way of Information Extraction. ACL 2022.

---

### Meta-Review · Area_Chair_gFEk · 2023-09-16

**Recommendation:** 3

**Metareview:**

The paper presents a prompt-based approach to IE where users extract tables of text from documents by providing instructions.  A Llama-7B model is fine-tuned on a dataset of instructions, input documents and output tables generated by ChatGPT.  The approach that is taken seems to be similar to Alpaca, however the model is specifically targeting IE based on user instructions.

The paper is well written, and the experiments seem convincing.  Reviewers raised some concerns, including the motivation for the new task and regarding the chosen evaluation metrics.

---

### Decision · Program_Chairs · 2023-10-07

**Decision:**

Accept-Main

**Comment:**

The paper presents a prompt-based approach to IE where users extract tables of text from documents by providing instructions.  A Llama-7B model is fine-tuned on a dataset of instructions, input documents and output tables generated by ChatGPT.  The approach that is taken seems to be similar to Alpaca, however the model is specifically targeting IE based on user instructions.

The paper is well written, and the experiments seem convincing.  Reviewers raised some concerns, including the motivation for the new task and regarding the chosen evaluation metrics.